# Non-catalytic motor domains enable processive movement and functional diversification of the kinesin-14 Kar3

Christine Mieck[1†], Maxim I Molodtsov[1,2,4†], Katarzyna Drzewicka[1], Babet van der Vaart[1], Gabriele Litos[1], Gerald Schmauss[1,3], Alipasha Vaziri[1,2,4*], Stefan Westermann[1*]

[1]Research Institute of Molecular Pathology, Vienna, Austria; [2]Max F Perutz Laboratories, University of Vienna, Vienna, Austria; [3]Institute of Molecular Biotechnology, Vienna, Austria; [4]Research Platform Quantum Phenomena and Nanoscale Biological Systems, University of Vienna, Vienna, Austria

**Abstract** Motor proteins of the conserved kinesin-14 family have important roles in mitotic spindle organization and chromosome segregation. Previous studies have indicated that kinesin-14 motors are non-processive enzymes, working in the context of multi-motor ensembles that collectively organize microtubule networks. In this study, we show that the yeast kinesin-14 Kar3 generates processive movement as a heterodimer with the non-motor proteins Cik1 or Vik1. By analyzing the single-molecule properties of engineered motors, we demonstrate that the non-catalytic domain has a key role in the motility mechanism by acting as a 'foothold' that allows Kar3 to bias translocation towards the minus end. This mechanism rivals the speed and run length of conventional motors, can support transport of the Ndc80 complex in vitro and is critical for Kar3 function in vivo. Our findings provide an example for a non-conventional translocation mechanism and can explain how Kar3 substitutes for key functions of Dynein in the yeast nucleus.

*For correspondence: alipasha. vaziri@univie.ac.at (AV); westermann@imp.ac.at (SW)

†These authors contributed equally to this work

Competing interests: The authors declare that no competing interests exist.

## Introduction

Motors of the kinesin family are ubiquitous enzymes essential for intracellular transport along microtubules in eukaryotes. The mechanism by which kinesin motor proteins convert the chemical energy of ATP hydrolysis into coordinated, long-range directional movement has fascinated cell biologists, biochemists, and engineers for many decades. Biophysical studies of kinesins have focused on conventional Kinesin-1 and established the 'hand-over-hand' model for the processive walking behavior of this type of motor (*Asbury et al., 2003*; *Yildiz et al., 2004*; *Kaseda et al., 2003*). In analogy to other enzymes, the term 'processivity' describes the ability of individual motor molecules to undergo multiple catalytic cycles—and therefore translocate—before releasing from the microtubule.

Kinesin-14 family members, exemplified by the *Drosophila* Ncd motor, are common examples for nonprocessive kinesins (*Case et al., 1997*; *Foster and Gilbert, 2000*). They generate motility through the minus-end-directed rotational movement of a coiled-coil mechanical element that occurs upon ATP binding (*Endres et al., 2006*). After each catalytic cycle, Ncd motors release from the microtubule lattice, meaning that to support microtubule sliding and crosslinking in the spindle, many Ncd motors must work together cooperatively in an ensemble (*Braun et al., 2009*; *Fink et al., 2009*). Budding yeast kinesin-14 Kar3 is distinct from other family members in its heterodimeric composition with either Cik1 or Vik1 (*Manning et al., 1999*) (*Figure 1A*). High-resolution structural analysis has shown that these accessory proteins contain a motor homology domain that harbors a microtubule binding site but lacks the structural elements required to bind and hydrolyze ATP (*Allingham et al., 2007*). Biochemical experiments have

**eLife digest** Molecules can be transported around a cell by so-called motor proteins that move along a network of filaments called microtubules. Many motor proteins—including the kinesin family of these proteins—can only move in one direction along a microtubule. In most cells, kinesins tend to transport other molecules away from the center and towards the cell edge.

Kinesins can have different structures, but most are made up of two subunits that are joined and work together to create a walking-like movement. Each subunit has a region called a motor domain (also known as its 'head') that can bind to the microtubule and to a molecule called ATP, which provides the energy required for the motor to step forward.

Kinesins can be classed either as processive or non-processive motors. Processive motors can walk continuously along a microtubule for several hundred steps, whereas non-processive motors fall off after just a few steps. A motor protein called Kar3 belongs to a group of non-processive kinesins. Kar3 is unusual; unlike most of the motors in this group (which work together in pairs), Kar3 motor protein subunits each bind to and work with non-motor protein subunits, including one called Cik1. The head of the non-motor protein cannot bind to ATP, although it can bind to microtubules. This means that the non-motor protein subunits are not provided with the energy to make a stepping motion; this raises questions about how the Kar3 motor protein moves along the microtubule, and whether this affects the roles the motor performs.

Mieck et al. studied how a molecular motor made up of Kar3 and Cik1 moves along microtubules using sensitive microscopy that allows single molecules to be observed. This revealed that, contrary to what is expected from a non-processive motor, Kar3–Cik1 moves long distances on microtubules without detaching from them. Further investigation showed that Cik1 provides a 'foothold' for the motor, binding it to the microtubule in such a way that allows it to move along the microtubule in the opposite direction to most kinesins. In addition, Mieck et al. found that the Kar3–Cik1 motor binds to and transports a protein complex that is crucial for separating chromosomes during cell division.

A challenge for the future is to understand in even greater detail how the movement of such a motor occurs. If it doesn't 'walk' like other motors, then how can its motion be explained? The benefits for the cell that underlie why this mechanism evolved also remain to be discovered.

indicated that Cik1 and Vik1 modulate the interaction of Kar3 with microtubules (*Allingham et al., 2007*; *Chen et al., 2011*; *Rank et al., 2012*). In vivo Kar3's heterodimeric composition governs its subcellular localization and function: Kar3 in complex with Vik1 crosslinks parallel microtubules in proximity to spindle pole bodies during mitosis (*Manning et al., 1999*), whereas antiparallel microtubule sliding is powered by Cik1–Kar3 complexes that associate with the microtubule lattice and plus-ends during mitotic and meiotic events (*Maddox et al., 2003*; *Gardner et al., 2008*). In addition, Kar3 has been implicated in kinetochore capture and transport (*Middleton and Carbon, 1994*; *Tanaka et al., 2005*, *2007*). The unusual composition of the Kar3 motor with the combination of a catalytic and a non-catalytic domain, as well as its key roles for diverse cellular processes in yeast, has made it a particularly interesting object of study both from a biophysical and cell biological point of view. The understanding of the mechanistic basis of Kar3 function, however, has remained incomplete, as biochemical experiments have been limited to ensemble assays using truncated or artificially dimerized proteins. On the basis of such experiments and the interpretation of in vivo phenotypes, it has been proposed that Cik1–Kar3 acts as a microtubule depolymerase (*Chu et al., 2005*; *Sproul et al., 2005*; *Allingham et al., 2007*). We hypothesized that the presence of a non-catalytic domain may allow functionalities fundamentally different from conventional kinesin-14 homodimers. As the activity of individual full-length Kar3 motors had not been observed directly, we developed assays to investigate motors at the single molecule level and analyze the contribution of the non-catalytic domain.

## Results

### Cik1–Kar3 is a processive kinesin-14 motor with a single catalytic domain

To study Kar3 motors at the single molecule level, we developed a protocol to express and purify full-length kinesin-14 heterodimers from *Saccharomyces cerevisiae* using affinity tagged Cik1 and Kar3

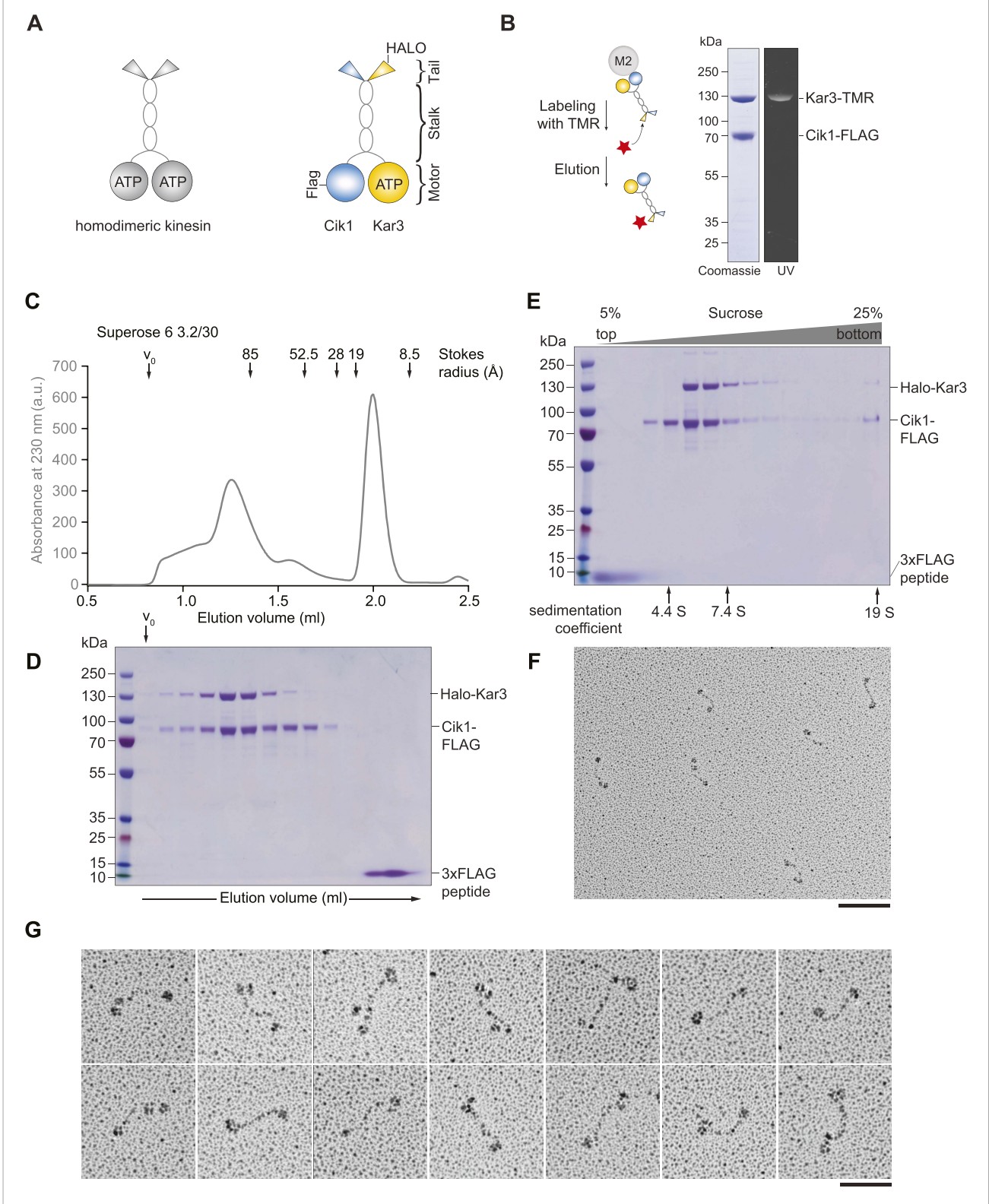

**Figure 1**. Purification and characterization of Cik1–Kar3 kinesin motors. (**A**) Schematic representation of conventional Kinesin-1 in comparison to the kinesin-14 Cik1–Kar3. (**B**) Purification of recombinant Cik1–Kar3 from yeast extracts. Motors are covalently labeled with Tetramethylrhodamine (TMR) via a HaloTag on the amino-terminus of Kar3. Coomassie-stained SDS-PAGE shows purity of the motor preparation and fluorescent labeling of the Kar3 subunit. (**C**) Size-exclusion chromatography of Cik1–Kar3-Halo motors on a Superose 6 column. The void volume of the column ($V_0$) and the elution

*Figure 1. continued on next page*

Figure 1. Continued

position of standard proteins with their respective stokes radii is indicated. (D) SDS-PAGE analysis of Superose 6 fractions from (C). (E) Sucrose gradient centrifugation of Cik1–Kar3 motors. Consecutive fractions from top to bottom of a 5–25 (wt/vol)% sucrose gradient were analyzed by SDS-PAGE and Coomassie staining. The gradient positions of standard proteins are indicated together with their sedimentation coefficients. (F) Low angle Pt/C rotary shadowing electron microscopy of Cik1–Kar3 motors obtained after size exclusion chromatography. Overview of Cik1–Kar3 motors, scale bar 100 nm. (G) Gallery view of selected Cik1–Kar3 motors, scale bar 50 nm.

fused COOH-terminally to a HaloTag that served as covalent attachment site for the fluorescent dye tetramethylrhodamine (TMR). Purification and labeling yielded a homogenous preparation containing a heterodimer of Cik1 and TMR-labeled Kar3 (*Figure 1B*). During size exclusion chromatography, Cik1–Kar3 motors eluted as a single major peak with a Stokes radius of ~9.1 nm, well separated from the void volume of the column (*Figure 1C,D*). Sucrose-gradient centrifugation revealed the presence of a single major species with an apparent sedimentation coefficient of ~5.6S (*Figure 1E*). Combining these hydrodynamic values yielded a native molecular weight of 214 kDa, close to the calculated molecular weight of a Halo-tagged Cik1–Kar3 heterodimer of 190 kDa. We further characterized the oligomeric state of full-length Cik1–Kar3 motors by performing low-angle Pt/C rotary shadowing electron microscopy on peak fractions from the gel filtration experiments. This analysis revealed individual well-defined, highly elongated molecules that were characterized by globular domains separated by a 61 ± 8 nm (mean ± SD, n = 100) long coiled-coil (*Figure 1F*). Typically, two closely spaced globular domains likely corresponding to catalytic and non-catalytic head domains were discernible at one end, while a single globular domain decorated the other (*Figure 1G*). The highly elongated shape of the Cik1–Kar3 molecules can explain their early elution from the gel filtration column. Overall, the hydrodynamic analysis and the direct visualization of motor molecules by EM support the presence of heterodimeric Cik1–Kar3 molecules.

We next observed the behavior of single motor molecules on surface-immobilized microtubules in vitro using time-lapse multi-color TIRF microscopy. Unexpectedly, and contrary to the classification of kinesin-14s as non-processive motors, Kar3 molecules displayed efficient ATP-dependent movement over several micrometers and accumulated at microtubule minus ends (*Figure 2A*, *Video 1*). Automated tracking of motors revealed a Gaussian velocity distribution of Cik1–Kar3 with a mean speed of 77 ± 23 nm/s (mean ± SD; *Figure 2B*) in range with reported microtubule gliding velocities for truncated Cik1–Kar3 molecules (*Chu et al., 2005*; *Allingham et al., 2007*). The motor is therefore approximately 10-fold slower than conventional Kinesin-1, but similar in speed to yeast cytoplasmic Dynein, the major minus-end directed-motor in eukaryotes (*Reck-Peterson et al., 2006*). The run-length histogram followed an exponential distribution and revealed that individual Cik1–Kar3 motors advanced processively for an average of 5.2 μm before detaching from the microtubule track (*Figure 2C*). The motile parameters were highly sensitive to the ionic strength of the imaging buffer: at the same ATP concentration higher salt concentration increased the mean squared displacement of the motors (*Figure 2D*), but decreased the on-rate, run length and minus-end dwell time (*Figure 2E*). We additionally noticed that Cik1–Kar3 oscillated back-and-forth when entering microtubule overlap zones. Because Cik1–Kar3 is exclusively moving towards the minus-end on single MT filaments, we concluded that reversing motors encountered an antiparallel-oriented MT bundle. Individual motors were able to switch the track microtubule multiple times leading to a prolonged association with antiparallel bundles (*Figure 2F*, *Video 2*).

## Individual Cik1–Kar3 heterodimers are sufficient for processive movement

The processive movement of Kar3 may either be a property of individual heterodimers or alternatively require the formation of motor ensembles combining multiple catalytic domains that coordinate stepping. Photobleaching experiments in the absence of nucleotide, in which the motor is persistently bound to the microtubule, revealed that 43 out of 50 molecules lost fluorescence in a single step consistent with the presence of a single Cik1–Kar3 heterodimer (*Figure 2—figure supplement 1A–D*). Mixing experiments combining TMR- with Alexa488 labeled Cik1–Kar3 molecules prior to imaging showed that the majority of moving complexes

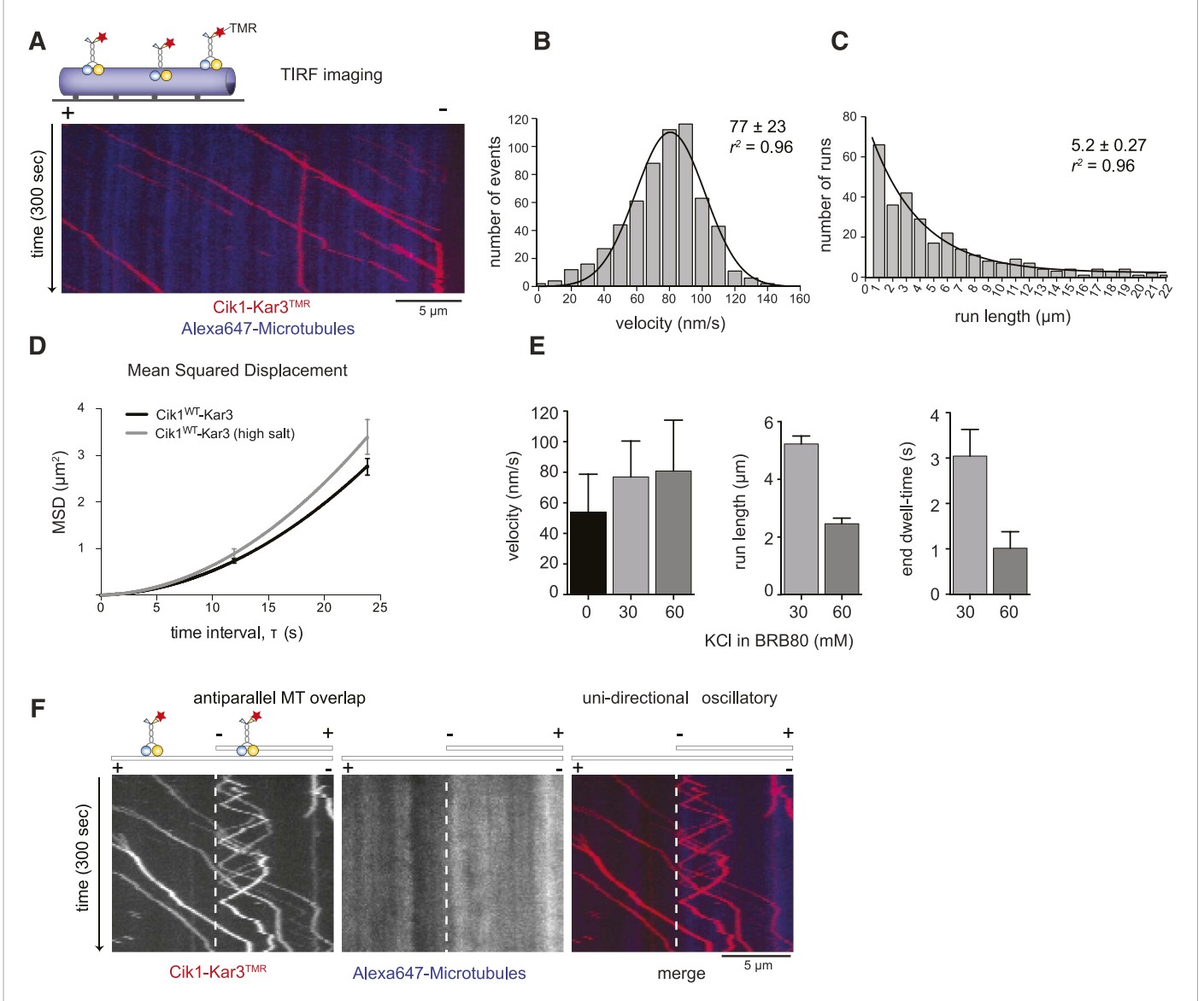

**Figure 2.** Cik1–Kar3 motors move processively with a single catalytic domain. (**A**) Kymograph showing two-color time lapse TIRF microscopy of Cik1–Kar3 (red) moving along taxol-stabilized microtubules (blue). See **Video 1** for example of Cik1–Kar3 motility. (**B**) Histogram of velocities of Cik1–Kar3 molecules moving along taxol-stabilized microtubules (fit with a Gaussian function, black line). The mean velocity is 77 ± 23 nm/s, n = 699. (**C**) Histogram of run lengths of Cik1–Kar3 molecules moving along taxol-stabilized microtubules, n = 209. (**D**) Mean-squared displacement analysis of wild-type Cik1–Kar3 at two different salt concentrations in the presence of ATP. (**E**) Influence of ionic strength on the motile properties of Cik1–Kar3 molecules. Experiments were performed in BRB80-based imaging buffer containing the indicated concentrations of KCl. (**F**) Behavior of Cik1–Kar3 in microtubule networks. Typical kymograph showing directional movement of Cik1–Kar3 on single microtubules vs repeated directionality switches of individual motors in antiparallel overlaps. The dashed line indicates the beginning of an overlap zone. See **Video 2**.

The following figure supplements are available for figure 2:

**Figure supplement 1**. Characterization of Cik1–Kar3 motility.

**Figure supplement 2**. A single Cik1–Kar3 heterodimer is sufficient to form a processive complex.

**Figure supplement 3**. Mathematical model of the cooperative kinesin movement.

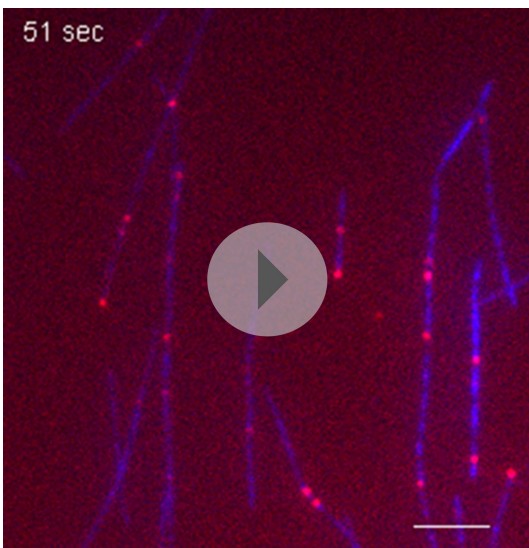

**Video 1.** Time-lapse two-color TIRF microscopy of Cik-Kar3-TMR motors (red) moving on taxol stabilized microtubules (blue). 100 frames were taken every 3 s. The video is played at 20 frames/s, scale bar: 5 µm. The video corresponds to *Figure 2A*.

**Video 2.** Behavior of Cik1–Kar3 in microtubule overlap zones. Two-color time-lapse TIRF video of Cik1–Kar3-TMR (red) moving on taxol-stabilized microtubules (blue). Note back and forth movement of individual Cik1–Kar3 motors in microtubule overlap zones. 100 frames were taken every 3 s, the video is played at 20 frames/s, scale bar 5 µm. The video corresponds to *Figure 2F*.

displayed exclusively either red or green fluorescence (*Figure 2—figure supplement 1E*, *Video 3*). To discriminate between processive and non-processive motility modes by an alternative approach, we varied the motor concentration in a standard microtubule gliding assay. Titration of Cik1–Kar3 over a 50-fold concentration range revealed no decrease in velocity for microtubule gliding, a characteristic feature of processive motility (*Hancock and Howard, 1998*) (*Figure 2—figure supplement 1F*). Contrary to previous reports, we did not observe substantial depolymerization of taxol-stabilized microtubules in the presence of Cik1–Kar3. On dynamic microtubules motors moved towards the minus-ends, overall microtubule dynamics appeared unchanged in the presence of Cik1–Kar3 and catastrophes did not coincide with plus-end localization of the motor (*Figure 2—figure supplement 1G*).

Quantification of TMR brightness of moving motors allowed us to compare the motile properties of Cik1–Kar3 heterodimers vs larger teams that consisted of two or more heterodimers (*Figure 2—figure supplement 2A–G*). We found that Cik1–Kar3 velocity was largely independent of motor team size (*Figure 2—figure supplement 2H,I*). The run length, however, increased with larger team size while the variance of the velocity decreased (*Figure 2—figure supplement 2J*). These motile behaviors of Cik1–Kar3 complexes of different sizes can be quantitatively explained by a biophysical model in which individual motors influence each other through mechanical coupling with spring-like properties (*Figure 2—figure supplement 3* and *Supplementary file 1*). Importantly, a key feature of the biophysical model is the ability of an individual Cik1–Kar3 heterodimer to move processively.

## Mutations in the non-catalytic head domain abolish directional movement

We next sought to establish the molecular requirements for Kar3 motility: processive movement could be a property intrinsic to the head domains or require secondary microtubule interaction sites in the motor tails (*Gudimchuk et al., 2013*; *Su et al., 2013*). To distinguish between these possibilities, truncation constructs were designed, systematically eliminating parts of the tail, coiled-coil, or globular domains of either the motor or the partner protein (*Figure 3A*). Truncation of the amino-terminal tail domain, either in Kar3 (up to aa 174) or in Cik1 (up to aa 250) allowed robust processive movement in the single molecule assay and also had little effect on multi-motor gliding velocity. This distinguishes Cik1–Kar3 from the homodimeric *Drosophila* kinesin-14, Ncd, which has been shown to be capable of a weakly

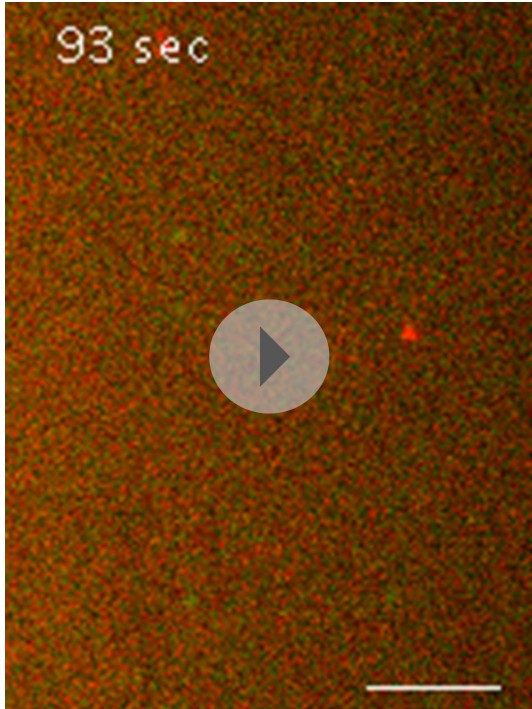

**Video 3.** Mixing experiment to demonstrate processivity of individual Cik1–Kar3 heterodimers. Cik1–Kar3-TMR motors (red) were mixed with Cik1–Kar3-Alexa488 motors (green) and imaged by multi-color TIRF microscopy on taxol-stabilized microtubules. Red and green motors are seen moving in opposite directions because of closely spaced microtubules with opposite orientation. Frames were taken every 3 s for 100 frames, the video is played at 20 frames/s. Scale bar corresponds to 5 μm. The video corresponds to *Figure 2—figure supplement 1E*.

processive motion at very low ionic strength depending on its tail region that acts as an electrostatic tether to microtubules (*Furuta and Toyoshima, 2008*). Further truncations of the coiled-coil interfered with heterodimer formation (not shown). In contrast, carboxyterminal truncations that either completely eliminated the predicted globular Cik1 motor homology domain (Cik1$^{1–360}$) or removed a portion of the carboxyterminus (Cik1$^{1–521}$) had severe effects and prevented directional movement. The specific nature of the defect was most apparent for the shorter truncation Cik1$^{1–521}$–Kar3, which was able to bind microtubules under our standard conditions, but instead of smooth translocation it displayed erratic forward and backward displacements that did not lead to directional movement as revealed by kymographs (*Figure 3B*, *Videos 4 and 5*). The defect imposed by the Cik1$^{1–521}$ mutation was also readily apparent in multi-motor gliding assays, where microtubules frequently switched their direction of movement and displayed little net transport (*Figure 3C*). The pronounced defect of the Cik1$^{1–521}$ mutant points to an essential role for the non-catalytic head in the motility mechanism. To corroborate this point, we also co-overexpressed Flag-tagged Kar3 with Kar3-Halo and purified fluorescently labeled Kar3 homodimers that can form in the absence of Cik1 (*Chu et al., 2005*). We failed to observe processive movement of Kar3-Kar3 combining two catalytic domains or of Kar3-Kar3$^{rigor}$ complexes combining catalytically active and inactive Kar3 heads (*Figure 3A*). These results point to a specific role of the non-catalytic Cik1 head that cannot be simply replaced by a second catalytically active or inactive Kar3 head.

## The non-catalytic domain restrains diffusion of Kar3 motors

To analyze the motile cycle in more detail, we imaged single Cik1–Kar3 motors in different nucleotide states by TIRF microscopy with high temporal resolution. In the absence of nucleotide (+Apyrase) and in the presence of the non-hydrolyzable ATP analog AMP-PNP Kar3 motors bound with high affinity to the microtubule but displayed no movement (*Figure 4A*). Binding under these conditions was sensitive to ionic strength with on-rate and lifetime of motor–microtubule interactions decreasing as the salt concentration was raised from 30 mM to 100 mM KCl (not shown). In the ADP state, motors displayed diffusive microtubule interactions (D = 0.061 ± 0.003 μm$^2$/s) with short residence times (τ = 0.6 ± 0.1 s) (*Figure 4B,C*).

What is the contribution of the non-catalytic domain to the translocation mechanism? To gain insights into this question, we quantitatively compared the microtubule-binding properties of single wild-type Cik1–Kar3 molecules with a mutant that lacks the globular Cik1 motor homology domain (Cik1$^{1–360}$–Kar3) (*Figure 4D*). In contrast to wild-type motors, the Cik1$^{1–360}$–Kar3 mutant displayed short-lived diffusive interactions with microtubules (*Figure 4D*). MSD data obtained in ATP for wild-type and mutant were fitted to the equation $\langle x^2 \rangle = a \cdot t^n$. n = 1 suggests standard diffusion and in this case, a = 2D, where D is the one-dimensional diffusion coefficient. For the mutant, an optimal fit was obtained with n = 1.01 ± 0.04, indicating unconstrained diffusion with no directional component. By contrast, n = 1.96 ± 0.03 was obtained for the wild-type motor, consistent with directional processive movement (*Figure 4E*). Further MSD analysis indicated that

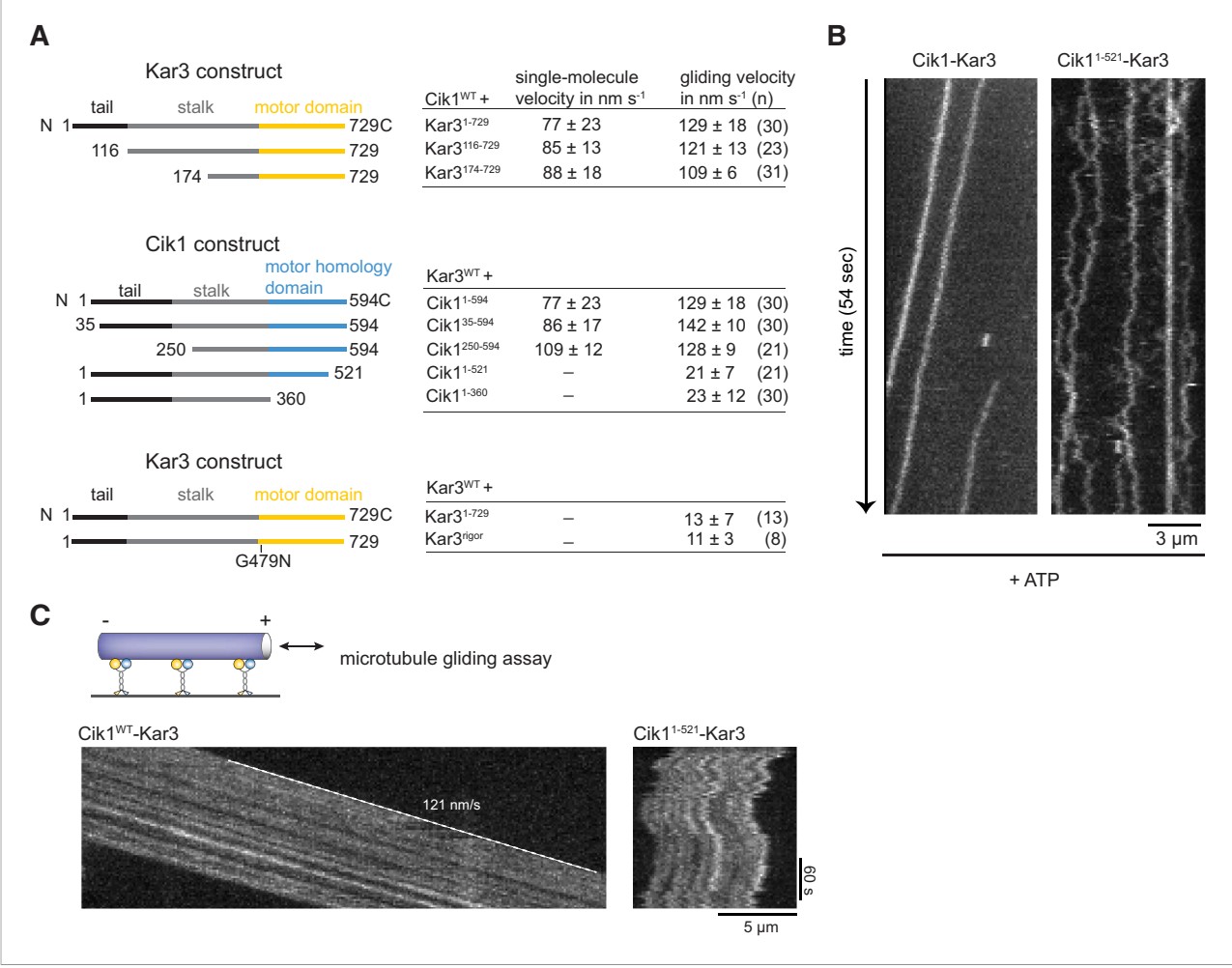

**Figure 3**. Molecular requirements for processivity and identification of a translocation-deficient Cik1 mutant. (**A**) Schematic showing analyzed Cik1 and Kar3 truncation constructs with the corresponding results from TIRF assays and multi-motor gliding assays. All constructs contained the Halo-tag at the aminoterminus of Kar3 for fluorescent labeling with TMR. (**B**) Kymographs of TMR-labeled Kar3 complexes containing either full-length Cik1 (aa 1–594) or the carboxyterminal truncation mutant Cik1[1–521]. See *Videos 4 and 5*. (**C**) Kymograph of microtubule-gliding by full-length Cik1–Kar3 and the Cik1[1–521]-Kar3 mutant. Note the impairment of directional translocation by the Cik1[1–521]–Kar3 mutant.

the mutant is about 20-fold more diffusive than the wild-type motor in the no nucleotide state ($D = 0.01 \pm 0.0006$ $\mu m^2/s$ vs $0.00045 \pm 0.00011$ $\mu m^2/s$) and about 30-fold more diffusive in the AMPPNP state ($D = 0.0036 \pm 0.00011$ $\mu m^2/s$ vs $0.00011 \pm 0.00001$ $\mu m^2/s$) (*Figure 4F*). Further analysis indicated that the Cik1[1–360] mutation also increased diffusion in the ADP state, but the difference was less pronounced in comparison to the other states (*Figure 4G,H*). We additionally constructed and purified a monomeric Kar3 head encompassing residues 353–729 with an N-terminal Halo-Tag. Under standard imaging conditions in the presence of ATP only very short-lived microtubule interactions without directional movement could be observed (*Figure 4I*). Taken together, these results indicate only a heterodimer containing a non-catalytic Cik1 head is able to move processively and a key contribution of the non-catalytic domain is to promote tight, non-diffusive binding of the motor in no nucleotide and AMPPNP states.

## The non-catalytic head determines the velocity of the Kar3 motor

Having established that the non-catalytic partner is critical for achieving mechanically processive Kar3, we sought to gain further insights into the underlying mechanism by studying the alternative

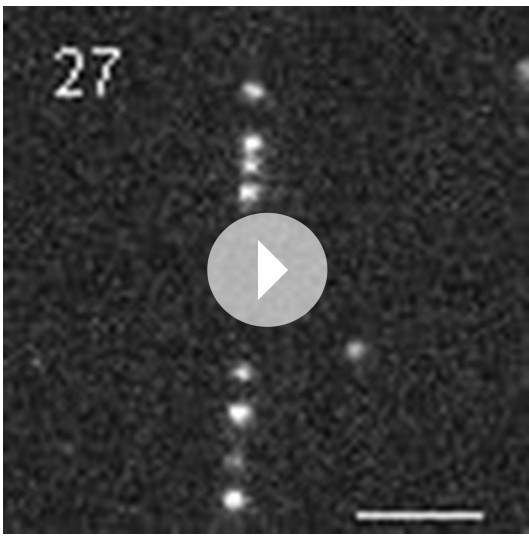

**Video 4.** Wild-type Cik1–Kar3-TMR in the presence of 5 mM ATP moving on taxol-stabilized microtubules imaged at higher frame rate in single color with TIRF microscopy. Time between frames is 273 ms, the frames are numbered in the upper left corner, total length of the video is 54 s. The video is played at 20 frames/s. Scale bar is 3 μm. The video corresponds to *Figure 3B*.

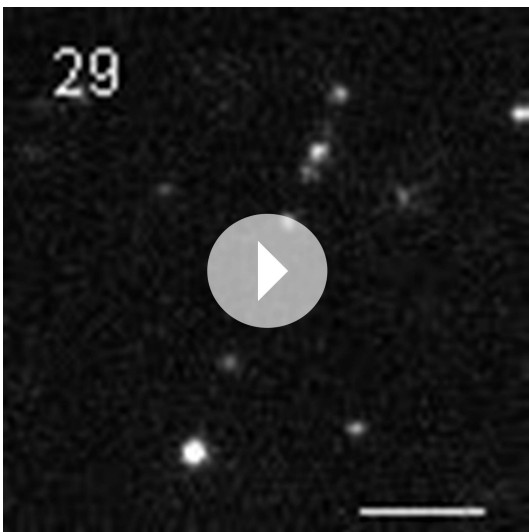

**Video 5.** Cik1–Kar3-TMR with truncation of the non-catalytic Cik1 globular domain (Cik1 1–521) interacting with taxol-stabilized microtubules in the presence of 5 mM ATP. Frame rate and imaging conditions as in *Video 5*. Note non-directional movement of Cik1¹⁻⁵²¹-Kar3. Scale bar 3 μm. The video corresponds to *Figure 3B*.

Vik1–Kar3 motor. The non-catalytic proteins Cik1 and Vik1 are paralogs that display about 45% sequence similarity between each other. We purified Vik1–Kar3 complexes from yeast extracts and imaged their interaction with microtubules under the same conditions as previously employed for Cik1–Kar3 (*Figure 5A*). Interestingly, Vik1–Kar3 also displayed highly processive movement, but with faster velocity compared to Cik1–Kar3. Under standard assay conditions, Vik1–Kar3 complexes moved at 234 ± 29 nm/s towards microtubule minus-ends where they strongly accumulated (*Figure 5B*, *Video 6*). To ask how these differences might be determined by the non-catalytic partner, we constructed a chimeric protein in which the globular motor homology domain and the neck of Vik1 (aa 351–647) were fused to the tail domain of Cik1 (aa 1–353) (*Figure 5C*). The Vik1–Cik1 chimera formed a stable heterodimer with Kar3-HALO and supported movement with 188 nm/s to the minus-end, significantly faster than Cik1–Kar3 and approaching the velocity of Vik1–Kar3 (*Figure 5D,E*). Thus, the globular non-catalytic domain is a key determinant for setting the velocity of the motor. The engineered chimeric protein was able to functionally replace Cik1 in cells as demonstrated by its ability to rescue the phenotypes of a Cik1 deletion and to support growth at wild type level under all tested conditions (*Figure 5F*).

## Differential binding partners determine the subcellular localization of Kar3 in vivo

In addition to directly controlling the motile characteristics of individual Kar3 motors, the non-catalytic partners may also provide functional diversification by allowing differential interactions with regulatory proteins or cargos. To identify such interactors, we performed affinity purifications of Cik1–FLAG at different cell cycle stages and analyzed associated proteins by mass spectrometry. In addition to Kar3, we reproducibly mapped two plus-end tracking proteins Bim1 and Bik1, the EB1 and CLIP-170 homologues of budding yeast, as well as the microtubule-binding Ndc80 kinetochore complex (*Figure 6A*). By performing pull-down assays with recombinant proteins, we established that Cik1–Kar3, but not Vik1–Kar3, directly interacted with Bim1 (*Figure 6B*). The binding interface involved the C-terminal EB homology domain of Bim1 and the aminoterminal tail domain of Cik1 (*Figure 6—figure supplement 1A–D*). A physical interaction with Bim1 may account for the previously observed microtubule plus-end localization of Cik1–Kar3 in vivo (*Sproul et al., 2005*;

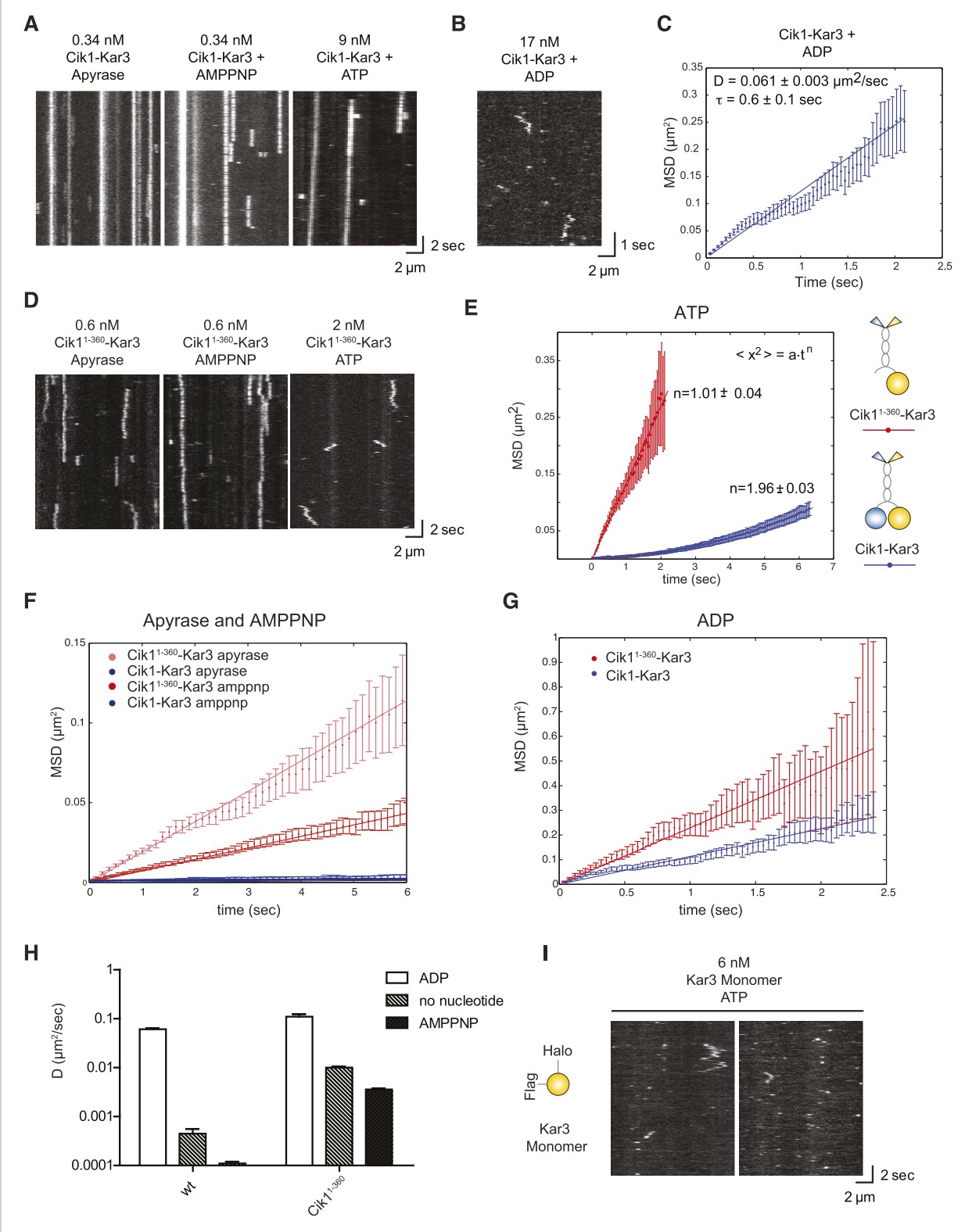

**Figure 4**. Single-molecule analysis of Kar3 motors in different nucleotide states. (**A**) Kymographs showing single molecule TIRF microscopy of Cik1–Kar3 motors in different nucleotide states. Videos were taken with high temporal resolution (35 frames per second). Note the different motor concentrations and the directional displacement of Cik1–Kar3 molecules in the presence of ATP. (**B**) Diffusive movement of individual Cik1–Kar3 molecules in the presence of ADP. (**C**) MSD analysis of the diffusive movement of Cik1–Kar3 motors in the presence of ADP. Molecules with lifetimes between 0.5 and 5 s

*Figure 4. continued on next page*

Figure 4. Continued

were analyzed. (**D**) Typical kymographs of Cik1$^{1-360}$–Kar3 motors lacking the non-catalytic head domain in no-nucleotide (Apyrase), AMPPNP and ATP states. Note the diffusive interactions of the motor with the microtubule in comparison to 3A. (**E**) Mean-squared displacement (MSD) analysis of Cik1–Kar3 and Cik1$^{1-360}$–Kar3 motors in the presence of ATP. Data points were fitted to the formula $<x^2> = a \cdot t^n$. n = 1 for Cik1$^{1-360}$–Kar3 indicates random diffusion without bias and constraints, while n = 2 for the wild-type motor indicates directional, processive movement. (**F**) Quantitative comparison of microtubule interactions of wild-type vs Cik1$^{1-360}$–Kar3 motors in no nucleotide and AMPPNP states by mean squared displacement analysis. (**G**) MSD analysis of wild-type vs Cik1$^{1-360}$–Kar3 motors in the ADP state. (**H**) Summary of the diffusion coefficients obtained for wild-type vs Cik1$^{1-360}$-Kar3 motors in three different nucleotide states. Note the logarithmic scale on the y-axis. (**I**) Typical kymographs of a monomeric Kar3 head construct (residues 353–729 fused to an N-terminal Halo tag) in the presence of ATP.

*Gardner et al., 2008*). Live cell imaging showed that the localization of Kar3 to distinct foci along the yeast spindle, as well as to the tips of shmoo-tip-directed microtubules in alpha-factor-arrested cells (*Figure 6—figure supplement 1E*), was abolished in a *bim1Δ* strain, pheno-copying a *cik1* deletion (*Manning et al., 1999*) (*Figure 6C,D,E*). Consistent with our biochemical experiments, spindle localization was maintained in the Cik1$^{1-360}$ and Cik1$^{1-521}$ mutants. Thus, the Cik1 tail domain specifies the differential localization of Kar3 in vivo by allowing a direct interaction with Bim1.

The insights into the localization determinants of the Kar3 motor allowed us to evaluate the key roles of Kar3 in the cell: yeast strains expressing Cik1$^{1-360}$ or Cik1$^{1-521}$ from the endogenous chromosomal locus displayed slow growth at 25°C and were inviable at 37°C, similar to a *cik1* deletion (*Figure 6F*). By contrast neither a *bim1* deletion, which eliminates spindle localization, nor a *vik1* deletion, removing spindle pole localization, displayed a pronounced growth phenotype (data not shown). Together with our finding that the Vik1$^{head}$–Cik1$^{tail}$ chimera fully supports viability, this implies that the key function of Kar3 in vegetatively growing yeast cells requires motility—supported by any of the two non-catalytic heads—and the Cik1 tail domain, but does not involve spindle localization via Bim1. Based on previously characterized mutants and on our finding that the Ndc80 complex co-purifies with Cik1, this supports a key role for Kar3 at the kinetochore. Consistent with this notion, we found that 80% of Cik1$^{1-521}$ cells arrested as large budded cells upon shift to the restrictive temperature, indicative of mitotic checkpoint activation. Analysis of chromosome segregation at the semi-permissive temperature of 25°C by fluorescent labeling of chromosome V showed that 50% of large budded Cik1 mutant cells attempted nuclear division without proper bi-orientation of sister chromatids (*Figure 6G*). We conclude that the key contribution of the non-conventional Cik1 motility mechanism for cell proliferation lies in the promotion of sister chromatid bi-orientation in mitosis.

## Cik1–Kar3 motors can promote transport of the Ndc80 kinetochore complex in vitro

To ask whether Cik–Kar3 can act as a transport motor, we first performed bulk in vitro recruitment assays in the absence of nucleotide using purified motor proteins and recombinant, fluorescently labeled Ndc80 complex as the candidate kinetochore binding partner revealed in the mass spectrometry experiments. In the absence of motor, the Ndc80 complex only weakly associated with taxol-stabilized microtubules under standard conditions (*Figure 7A,B*). Vik1–Kar3 motors strongly decorated microtubules in the absence of ATP, but had no effect on the Ndc80 complex. By contrast, combining Cik1–Kar3 and the Ndc80 complex resulted in decoration of microtubules with both molecules. Upon lowering protein concentrations, Cik1–Kar3 and the Ndc80 complex co-localized to distinct spots on the microtubule where their intensities were correlated (*Figure 7C*). Inclusion of ATP initiated co-transport of the Ndc80 complex and Cik1–Kar3 towards the minus end (*Figure 7D*, *Video 7*). While transporting the Ndc80 complex, the average speed of the motor was 76 ± 15 nm/s (mean ± SD), indicating that cargo binding did not significantly impede movement. The value is also in range with the reported velocity for kinetochore transport in vivo (*Tanaka et al., 2005*). Microtubule binding by the Ndc80 complex was not required for efficient transport, as shown by effective recruitment and translocation of the microtubule-binding defective calponin-homology domain mutant K122E K204E (*Lampert et al., 2013*) (*Figure 7—figure supplement 1*). This result is in agreement with the observation that Kar3 localizes to unattached kinetochores prior to microtubule binding. We conclude that the Ndc80 complex can be transported by the Kar3 motor in vitro and that in addition to promoting processive motility a key function of Cik1 may lie in the differential binding of this complex.

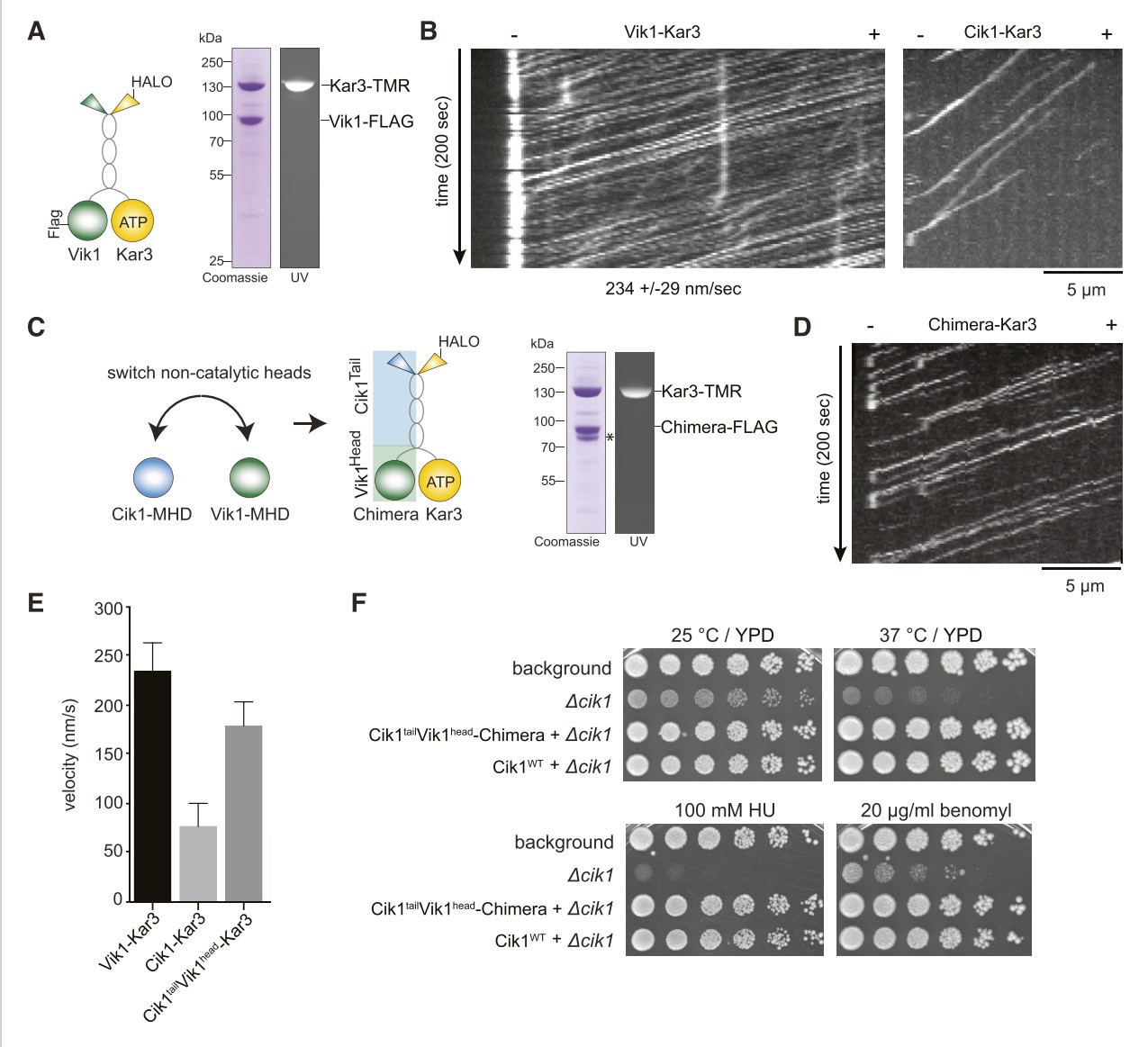

**Figure 5**. The non-catalytic head domain determines the velocity of the Kar3 motor. (**A**) Schematic representation of the kinesin-14 Vik1–Kar3 and purification of recombinant Vik1–Kar3 from yeast extracts. Motors are covalently labeled with Tetramethylrhodamine (TMR) via a HaloTag on the amino-terminus of Kar3. Coomassie-stained SDS-PAGE shows homogeneity of the motor preparation and fluorescent labeling of the Kar3 subunit. (**B**) Kymographs of single color time lapse TIRF microscopy highlighting the velocity difference between Vik1–Kar3 and Cik1–Kar3 when moving along taxol-stabilized microtubules. (**C**) Construction and purification of a chimeric motor in which the non-catalytic heads were switched. An asterisk denotes a proteolytic degradation product. (**D**) Kymograph of typical single molecule runs by the chimeric Cik1[Tail]–Vik1[Head] motor. (**E**) Comparison of velocities of the different Kar3 constructs. (**F**) Serial dilution growth assay of the indicated yeast strains at different temperatures and conditions. Note that the integration of the chimeric Cik1[Tail]–Vik1[Head] fusion protein fully rescues the phenotypes of a Cik1 deletion.

## Discussion

### Kar3 motors and non-conventional translocation along microtubules

Our study provides direct evidence for the processivity of Kinesin-14 motors by analyzing full-length yeast Kar3 motors on the single-molecule level. Contrary to other kinesin-14s, we demonstrate that individual Cik1–Kar3 motors can move processively towards the minus-end and show that the non-catalytic Cik1 head domain is functionally required for this activity. We further demonstrate that the combination of Kar3 with two different non-catalytic partners generates motors with different motile

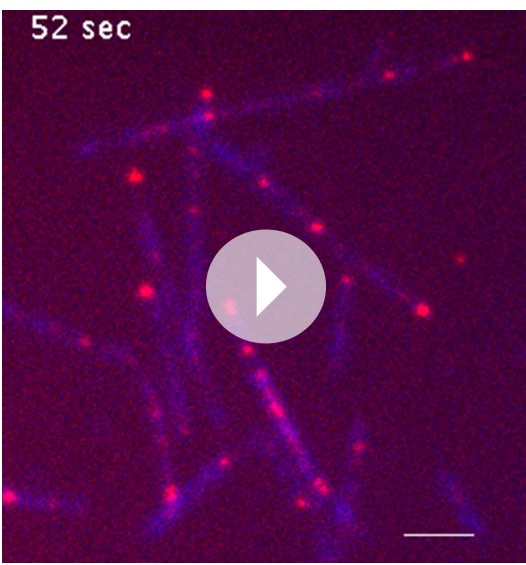

**Video 6.** Vik1–Kar3 motors display processive movement and pronounced minus-end dwelling. Two-color time lapse TIRF microscopy of Vik1-Kar3-TMR (red) on taxol-stabilized microtubules (blue). Note strong accumulation of motors at minus-ends at the end of the video. 100 frames were taken every 3 s, video is played at 20 frames/s, scale bar is 5 μm. The video corresponds to *Figure 5B*.

properties and also allows the differential binding of partner proteins such as Bim1 and the Ndc80 complex (*Figure 8A*).

The demonstration of Cik1–Kar3 processivity presented here provides evidence that effective translocation along microtubules does not strictly require the conventional hand-over-hand walking mechanism that has been established for conventional Kinesin-1. So far this point has been most clearly demonstrated for the Dynein motor, as recent studies have shown that cytoplasmic dynein is capable of processive movement in vitro despite inactivating mutations in one of the two force-generating AAA ring domains (*DeWitt et al., 2012*; *Qiu et al., 2012*), or even as a combination of one active head with a second passive microtubule-binding domain (*Cleary et al., 2014*).

Also several kinesin motors have been reported to move in violation of the hand-over-hand model: monomeric constructs of the kinesin-3 KIF1A have been shown to move along microtubules in vitro using a biased diffusion mechanism (*Okada and Hirokawa, 1999*), but the in vivo function of these motors probably involves dimeric molecules for effective cargo transport (*Tomishige et al., 2002*; *Endres et al., 2006*). We could not detect evidence for processive movement of monomeric Kar3, making it unlikely that it works via a KIF1A-type mechanism. Among the kinesin-14 family, Ncd can exhibit processive motility but requires at least two homodimers coupled together (*Furuta et al., 2013*). A number of studies have shown that inactivating mutations in one of the two heads of conventional kinesin can still allow residual processivity (*Subramanian and Gelles, 2007*; *Thoresen and Gelles, 2008*; *Kaseda et al., 2003*). We note, however, that in these examples processive movement of the mutant heterodimeric kinesins was compromised compared to wild-type homodimers. By contrast, Kar3 does not simply tolerate the loss of a second active head but instead it has evolved a specific requirement for the non-catalytic domain in order to move processively. In other words, Kar3 can only walk with a combination of an active and an inactive 'leg'.

## A working model for the movement of Kar3 kinesins

A precise elucidation of the stepping mechanism of Kar3 motors will require further biophysical experiments. Based on the results obtained in this and in previous studies, we propose a working model that describes the non-catalytic head as a 'foothold' for Kar3. Based on recent structural data Kar3 and its partner non-catalytic head may be located on adjacent protofilaments (*Rank et al., 2012*), where the non-catalytic domain could contribute to hold the motor complex in place and prevent diffusion. During its ATP cycle, the motor goes through rounds of tightly and weakly bound states. When ADP is bound, the motor is in its weakly bound state which allows for one-dimensional diffusion along the microtubule lattice. On its own, movement in this state would lack directionality and an additional mechanism providing minus-end directed bias is required. We speculate that the presence of Cik1 in combination with the kinesin-14 typical stalk rotation upon ATP binding by Kar3 (*Endres et al., 2006*; *Gonzalez et al., 2013*) might provide such a mechanism either by shifting the overall position of the molecule or by allowing subsequent directed binding/unbinding (*Figure 7C*). Because of the combination of one foot on the track and a diffusive movement to the next binding site, we refer to this possible mechanism as the 'skateboard' model. This model can explain a number of our experimental observations: the effect of raising the ionic strength is that initially motor movement occurs with faster

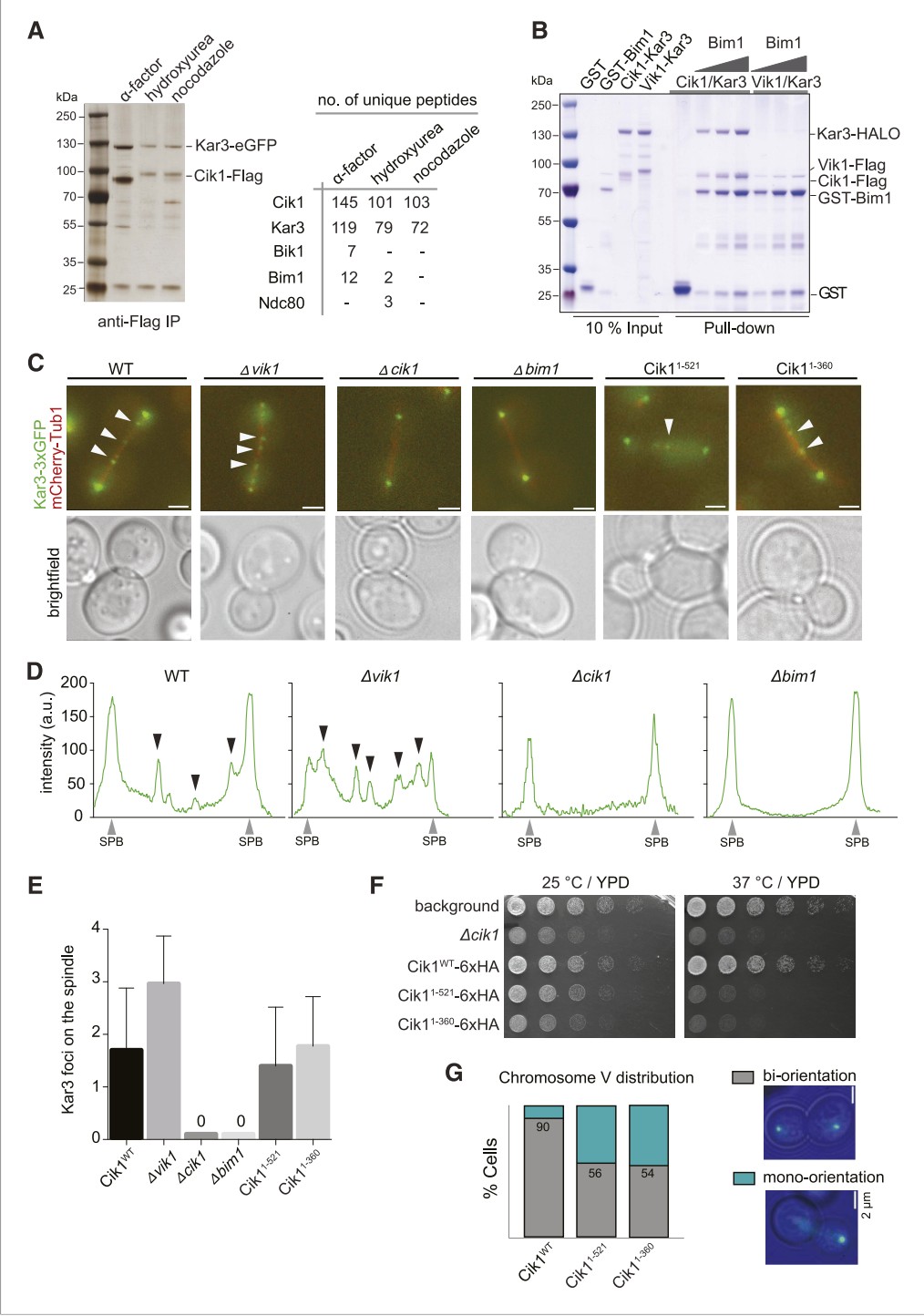

**Figure 6**. Cik1 specifies differential interactions of Kar3 motors to determine their subcellular localization. (**A**) Affinity purification of the Cik1–Kar3 complex from different cell cycle states and identification of interaction partners by mass spectrometry. (**B**) Pull-down assay with GST-Bim1 and Cik1- or Vik1-Kar3. Only Cik1-Kar3 interacts with Bim1 in a dose-dependent manner. (**C**) Localization of Kar3 in different deletion mutants in yeast investigated by live cell microscopy. (**D**) Panel shows line scans of Kar3–GFP intensity along the spindle axis. Arrowheads point to Kar3 foci along the spindle. Scale bar: 3 μm. Both Cik1 and Bim1 are required for the localization to the anaphase spindle. (**E**) Quantification of Kar3 foci on anaphase spindles in the indicated yeast strains. Kar3 spindle signals were never detected in a *bim1Δ* or *cik1Δ* strain. (**F**) The Cik1[1–521] and Cik1[1–360] mutants elicit a temperature-sensitive phenotype in vivo. Growth assay was performed with serial dilutions of the indicated yeast strains at different temperature on
*Figure 6. continued on next page*

*Figure 6. Continued*

rich medium (YPD). (**G**) Quantification of chromosome segregation in the indicated yeast strains at 25°C, n = 100 for all strains analyzed. Right panel shows representative fluorescent micrographs with segregation of fluorescently labeled Chromosome V.

The following figure supplement is available for figure 6:

**Figure supplement 1**. Biochemical and genetic characterization of the Cik1-Kar3-Bim1 interaction.

velocity, probably by increasing the diffusion term. At some point however, stable binding of the non-catalytic domain is prevented and directional displacement is disrupted, similar to the situation in which the motor lacks the motor homology domain. In addition, diffusive movement in the ADP state and the observed large variation in movement rates are compatible with the model. The biochemical properties of the non-catalytic head binding to the microtubule will have profound effects on the motile characteristics as evidenced by the different properties of Cik1–Kar3 and Vik1–Kar3 motors.

One of the open questions regarding the mechanism of processivity is how the motor homology domain can provide 'footing' in no nucleotide and AMPPNP states yet allow effective diffusion in the ADP state. In agreement with previous work, this suggests that similar to conventional kinesins, a form of head-to-head communication must occur in Kar3 motors such that the catalytic head influences the microtubule affinity of the motor homology domain (*Allingham et al., 2007*; *Chen et al., 2011*; *Duan et al., 2012*; *Rank et al., 2012*). Such coordination may involve intramolecular strain communicated by mechanical elements between the heads as has been reported for conventional kinesin (*Yildiz et al., 2008*).

## Implications for Kar3 function in vivo

Based on our experiments and the previously reported rates of microtubule shortening in the presence of Cik1–Kar3, we consider it unlikely that Kar3 functions as a microtubule depolymerase in vivo. We note that none of the Kar3 functions poses a strict requirement for such an activity and indeed recent experiments analyzing Kar3 function during karyogamy support the view that its primary function is transport, not shortening (*Gibeaux et al., 2013*).

What are the implications of Kar3's processivity? The ability to move processively may be used in the cell to enrich Kar3 motors at minus-ends. In addition, Kar3 could work more effectively in small teams if it can undergo multiple catalytic cycles before releasing from the microtubule. This may be especially important during kinetochore transport, where Ndc80 complexes can provide only a limited number of binding sites for the motor. Kar3 heterodimers have probably evolved from conventional Kinesin14-homodimers that performed Ncd-like roles in spindle organization. We speculate that with the exclusion of dynein from the budding yeast nucleus additional functional requirements for minus-end directed motility in spindle assembly and kinetochore function arose that could be fulfilled with the 'invention' of the Cik1 and Vik1 non-catalytic domains. While commonly perceived as conferring a loss of functionality, the pseudo-motor domains have the ability to convert a kinesin-14 into a processive motor and additionally create functional diversity through allowing differential interactions with partners. Together this allows Kar3 to function as a processsive sliding and transport motor that substitutes for key roles of dynein in the yeast nucleus. Other examples for heterodimeric motors exist in the Kinesin-2 family with different subunits conferring distinct activities (*Brunnbauer et al., 2010*). Kar3 motors may therefore constitute an extreme example for such diversification strategies that could be more widely used in other motor families. More generally, we note that pseudoenzymes have emerging roles in other cellular contexts, for example in the form pseudo-GTPases during human kinetochore assembly (*Basilico et al., 2014*), or as pseudo-kinases and pseudo-phosphatases in cell signaling (*Boudeau et al., 2006*; *Tonks, 2009*). This may suggest that the use of catalytically inactive enzyme derivatives could be a more widespread strategy employed by the cell. The establishment of an in vitro assay for Ndc80 transport serves as a starting point for a detailed analysis of the function and regulation of

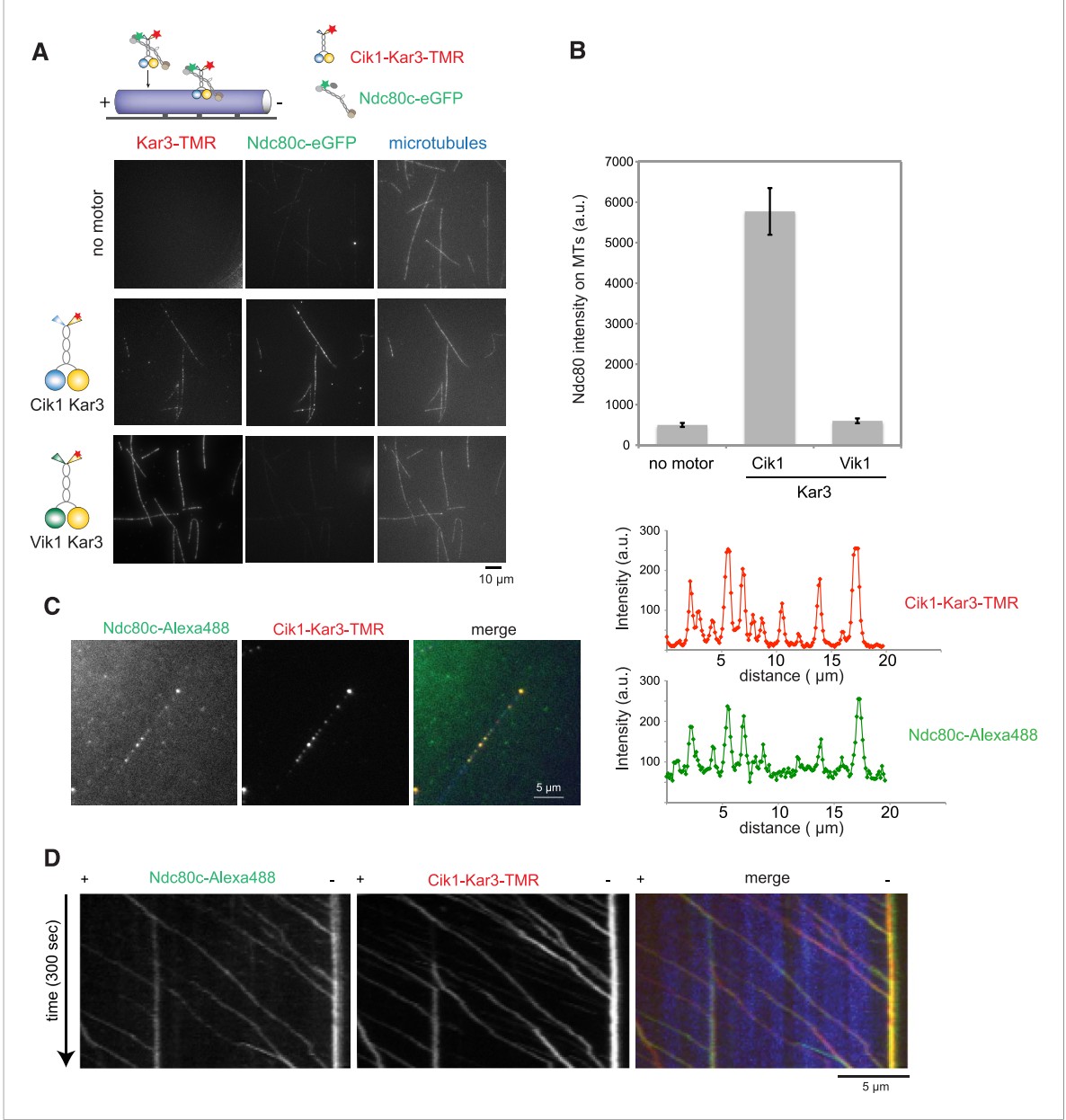

**Figure 7**. Processive transport of the Ndc80 kinetochore complex by Cik1–Kar3 motors in vitro. (**A**) Cik1–Kar3, but not Vik1–Kar3 recruits the Ndc80 kinetochore complex to taxol-stabilized microtubules in vitro. The experiment was performed with TMR-labeled Kar3 motors and fluorescently labeled recombinant Ndc80 complex in the absence of nucleotide. (**B**) Quantification of the recruitment experiment by analyzing fluorescence intensity of microtubule-bound Ndc80-eGFP complex in the presence of different motor constructs. Error bars denote standard error of the mean (s.e.m.) (**C**) Co-localization of the Ndc80 kinetochore complex and Cik1–Kar3 motors to distinct signals on the microtubule lattice. Right panel shows line scans of fluorescent intensity of Kar3 motors and Ndc80 complex along the length of a microtubule. (**D**) Kymograph showing triple-color time-lapse TIRF microscopy demonstrating that in the presence of ATP, Ndc80 complexes are processively transported towards microtubule minus ends by Cik1–Kar3 motors.

The following figure supplement is available for figure 7:

**Figure supplement 1**. Cik1–Kar3 transports a mutant Ndc80 kinetochore complex that is unable to bind MTs.

kinetochore motility. Furthermore, the unique design principle of the Kar3 motor, which allows the same catalytic domain to be paired with different non-catalytic heads, generates functional diversity that may also be exploited in nanotechnological applications.

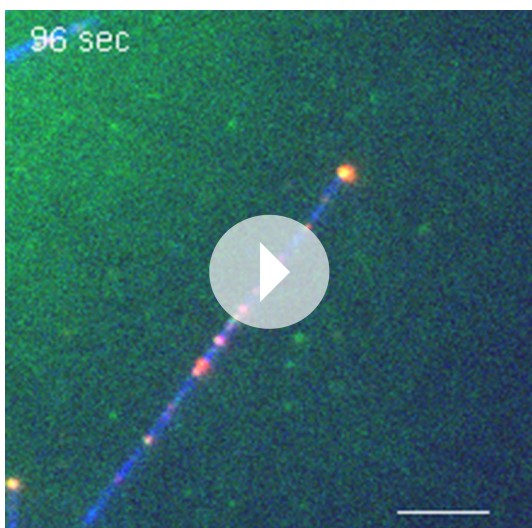

**Video 7.** Transport of the Ndc80 kinetochore complex by Cik1–Kar3 motors along taxol-stabilized microtubules. Triple-color TIRF video showing Ndc80-Alexa488 complex (green), Cik1–Kar3-TMR motors (red), and Alexa647-Taxol stabilized microtubules (blue) in the presence of 5 mM ATP. Frames were taken every 3 s. Note co-transport of Ndc80 complexes and Cik1–Kar3 complexes towards the end of the microtubule. Video is played at 20 frames/s, scale bar is 5 µm. The video corresponds to *Figure 7D*.

## Materials and methods

### Protein Biochemistry

The protein coding sequences of *S. cerevisae* (S288c) full-length Kar3, Cik1, and Vik1 were amplified by PCR and cloned into the over-expression pESC-TRP (Agilent Technologies, Santa Clara, CA) vector. The non-catalytic motor subunit was tagged C-terminally with 1xFLAG and the motor itself was fused to a HaloTag (DHA, Promega) at the 5′ end of the coding sequence, separated by a 13 amino acid linker. Mutated and truncated versions of Cik1 and Kar3 were generated by site-directed mutagenesis PCR (Phusion, Thermo Scientific). To determine the TMR-labeling efficiency, Halo-Kar3 was fused with enhanced green fluorescent protein (eGFP), creating Halo-eGFP-Kar3. All motor proteins were overexpressed in budding yeast using the pESC vectors (Agilent Technologies) following the manufacturer's instructions. In brief, yeast cells harboring the respective pESC plasmid were induced with 2% Galactose at $OD_{600}$ = 1.0 for 7–9 hr, harvested by centrifugation, washed, and frozen as droplets in liquid nitrogen. Lysis was conducted in liquid nitrogen using a freezer mill (Biospec Inc.). The cell powder was resuspended in lysis buffer (25 mM Hepes [7.4], 300 mM NaCl, 1 mM $MgCl_2$, 5% glycerole, 0.1 mM EDTA, 0.5 mM EGTA, 0.1% Tween-20, 0.01 mM ATP, 0.1 mM PMSF supplemented with PhosSTOP Phosphatase Inhibitor Cocktail [Roche]). The lysed cells were centrifuged twice, first at 43.146×g for 20 min and afterwards at 125.749×g for 1 hr. The resulting supernatant was incubated with M2 affinity agarose (Sigma–Aldrich) for 1 hr, gently rotating at 4°C. The agarose resin was washed five times with lysis buffer (adjusted to 150 mM NaCl, 0.09% Tween-20, omitting the PhosSTOP reagent). Elution of the kinesin heterodimeric constructs from M2 agarose was achieved by applying one resin volume of 3xFLAG peptide at final 2 mg/ml in lysis buffer (adjusted to 250 mM NaCl, 1 mM DTT, 0.09% Tween-20, omitting ATP and PhosSTOP). If needed, the elution was loaded onto a cation exchange chromatography (MonoS 5/50 GL, GE Healthcare) in running buffer (10 mM Hepes [pH 7.2], 150 mM NaCl, 1 mM $MgCl_2$, 1 mM DTT, 1 mM EGTA) to remove the FLAG peptide. Afterwards a linear salt gradient eluted a single peak, pure motor fraction at 250 mM salt. Elution fractions were supplemented with glycerol (final concentration: 5%), snap-frozen in liquid nitrogen and stored at −80°C. The protein concentration was measured using the DC Assay kit (Bio-Rad). All proteins were pre-cleared by centrifugation using a 0.1-µm spin filter (Millipore) to remove aggregates before each experiment.

Labeling of the $NH_2$-terminus of Kar3 with the HaloTag TMR ligand or HaloTag Alexa488 ligand (Promega, Madison, WI) was performed during the above-described purification before eluting the kinesin heterodimer from the M2 affinity agarose: the proteins were incubated with 10 µM HaloTag ligand for 3 hr. Extensive washing removed unbound TMR ligand and the kinesin was eluted as described before. In order to assess the labeling efficiency the Halo-eGFP-Kar3 construct was expressed, purified, and TMR-labeled. Observing the motor in our multi-color single-molecule imaging setup in the absence or presence of ATP revealed that >90% of eGFP-labeled kinesins also had a TMR-ligand covalently bound to the HaloTag.

To obtain a stoichiometric heterodimer of Cik1 and Kar3, the motor was subjected to analytical SEC. The purified motor was loaded onto a Superose 6 PC 3.2/30 column (GE Healthcare), and 100 µl fractions were collected and separated by SDS-PAGE. Proteins were stained with Coomassie Brilliant blue R250.

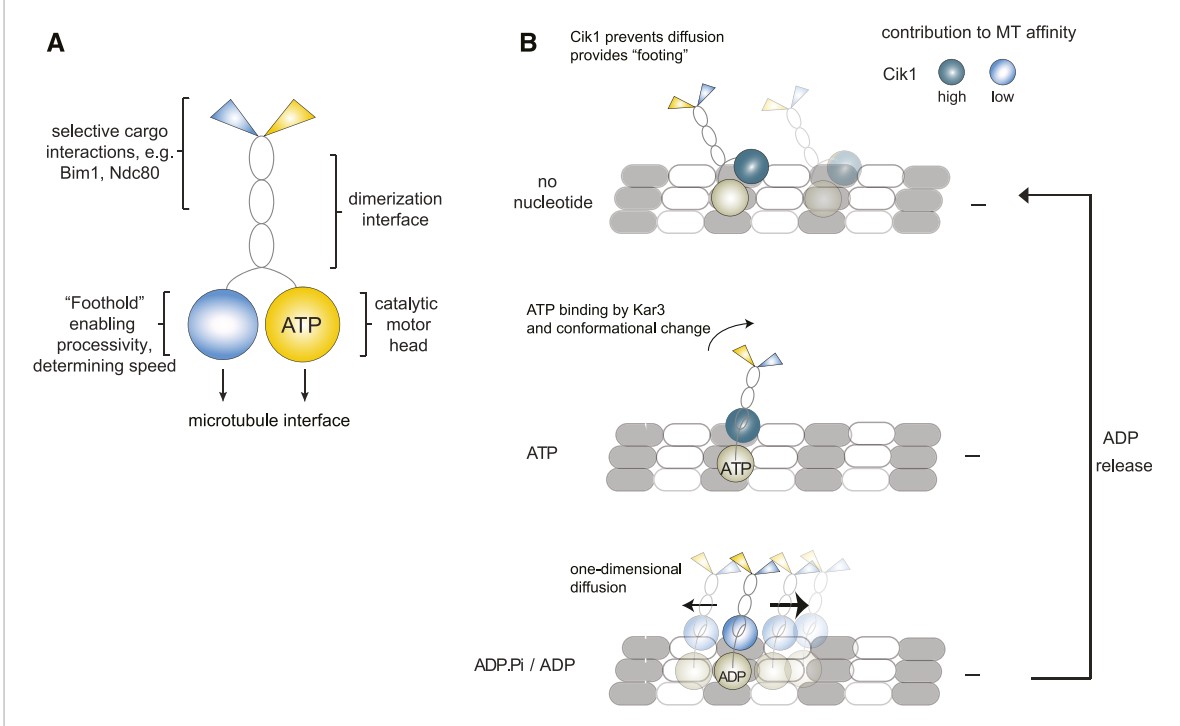

**Figure 8**. Model for Kar3 motility and function. (**A**) Functional contributions of the different domains of heterodimeric Kar3 motors analyzed in this study. (**B**) Model for Cik1–Kar3 motility depending on a non-catalytic head domain. Upon ATP uptake a conformational change occurs with the stalk rotating towards the minus end. A key role for the non-catalytic domain is to prevent diffusion in this state but allow one-dimensional diffusion in the subsequent ADP state with a bias towards the minus end.

Expression and purification of the full-length and mutant Ndc80 complex (Ndc80p-6xHis/Nuf2p-EGFP and Spc24p/6xHis/Spc25p) was performed as described previously (*Lampert et al., 2013*; *Lampert et al., 2010*). Bim1 was expressed and purified as described previously (*Zimniak et al., 2009*).

## Rotary shadowing electron microscopy of full-length Cik1–Kar3
Peak fractions of motor from the gel filtration experiments were diluted to a final concentration of 60 µg/ml in spraying buffer, containing 100 mM ammonium acatate and 30% (vol/vol) glycerol, pH adjusted to 7.4. After dilution, the samples were sprayed onto freshly cleaved mica chips and immediately transferred into a Bal-Tec MED020 high vacuum evaporator equipped with electron guns. After drying in the vacuum, the rotating samples were coated with 0.6 nm Platinum at an elevation angle between 5° and 6°, followed by 9.5 nm Carbon at 90°. The produced replicas were floated off from the mica chips and picked up on 400 mesh Cu/Pd grids. The grids were inspected in an FEI Morgagni 268D TEM operated at 80 kV. Electron micrographs were acquired using an 11 megapixel Morada CCD camera from Olympus-SIS. Images were examined and analyzed using ImageJ.

## Sucrose gradient centrifugation
Single-step purified Cik1–Halo–Kar3 was loaded on top of a 4.4 ml 5–25% linear sucrose gradient and spun at 50,000 rpm for 16 hr using an Sorvall TH-660 rotor and a Sorvall Discovery 90SE centrifuge. Fractions (270 µl) were collected and analyzed by SDS-PAGE and Coomassie Brilliant Blue R250 staining. The sedimentation value for Cik1-Halo-Kar3 was defined by comaring the mobilities of the motor with linear plots of mobility standards.

## Single molecule imaging assay
The single molecule motor assays were conceptually designed as described previously (*Korten et al., 2011*; *Lampert et al., 2010*; *Bieling et al., 2010*). Biotin-PEG-SVA- and mPEG-SVA-functionalized

coverslips (Laysan Bio) were prepared as described (*Lampert et al., 2010*; *Jain et al., 2012*). Coverslips were assembled onto passivated glass slides using double-sided tape creating a flow chamber. First, a solution of 1 mg/ml avidin DN (Vector Laboratories) was applied to the chamber for 30 min and exchanged for 1% pluronic F-127 (Sigma–Aldrich) in BRB80 (*Zimniak et al., 2009*; *Nitzsche et al., 2010*). Porcine-derived HiLyte-647-labeled, biotinylated and taxol-stabilized micro-tubules (MTs) were immobilized (labeled and biotinylated tubulin source: Cytoskeleton Inc., unlabeled tubulin was purified from pig brains as described previously [*Ashford and Hyman, 2006*]) and excess of MTs was washed out with BRB80 buffer supplemented with 5 μM taxol, 0.5% (vol/vol) β-mercaptoethanol, 4.5 μg/ml glucose, 200 μg/ml glucose-oxidase, and 35 μg/ml catalase. Single molecule imaging was performed by introducing the TMR- or Alexa488-labeled kinesins at low nanomolar range in assay buffer (BRB80, 0.33 mg/ml casein, 16.6 μM taxol, 0.13% [vol/vol] methylcellulose, 0.5% [vol/vol] β-mercaptoethanol, 4.5 μg/ml glucose, 200 μg/ml glucose-oxidase and 35 μg/ml catalase, 0.06% [vol/vol] Tween-20, the indicated amount of KCl and the respective amount of nucleotide) into the flow cell. Time-lapse videos were recorded at 28°C in 3 s intervals between frames (if not annotated differently) using a TIRF microscopy setup described previously (*Lampert et al., 2010*). Multi-color imaging was achieved by the use of an external filterwheel (Ludl Electronic Products Ltd.). Each channel (excitation: 488 nm, 561 nm, 639 nm) was exposed for 100 ms at every time-interval, recorded by a Cascade II EMCCD camera and projected to two-dimensional images (software: Metamorph [Molecular Devices], ImageJ). For photobleaching analysis, the oxygen-scavenger mix was omitted (β-mercaptoethanol, glucose, glucose-oxidase, catalase), and images were recorded at maximum laser power. Videos are represented as kymographs (time-space plot) or as example single frame (software: MetaMorph [Molecular Devices]).

High temporal resolution recordings (as presented in *Figure 2*) have been obtained using custom-made TIRF microscope based on Olympus IX-71 body and Coherent CUBE lasers in a temperature stabilized room (21 ± 0.1°C). Images were acquired using a Andor iXon3 897 EMCCD camera and subsequently analyzed using custom-made software written in MATLAB (MathWorks, Inc).

## Microtubule gliding assay

The microtubule gliding assay was designed as described before (*Nitzsche et al., 2010*). For this assay, hydrophobic coverslips were prepared according to the following scheme: sonication in acetone for 15 min was followed by sonication for 15 min in ethanol. Coverslips were incubated 1 hr in boiling Piranha solution and rinsed with a lot of water afterwards. Then rinsed with 0.1 M KOH, MilliQ and dried with nitrogen and immersed in 5% dichlorodimethylsilane in heptane for 1 hr at room temperature. Coverslips were rinsed again with MilliQ and sonicated for 5 min, sonicated in chloroform for 5 min and air-dried.

Motor proteins were attached to the hydrophobic coverslips via application of 0.2–20 μg/ml anti-Halo antibody (Promega) to the flow chamber. After the excess of antibody was washed out with BRB80 supplemented with 0.5 mg/ml casein, motors were introduced to the chamber and incubated for 5 min. Finally a microtubule-containing solution supplemented with 5 mM ATP and oxygen scavenger mix was perfused into the chamber. Microtubules used for this experiment were assembled from porcine tubulin mixed together with HiLyte-647-labeled tubulin (Cytoskeleton Inc.) in the presence of 10 μM taxol. Time-lapse videos were recorded at 28°C in 3-s intervals between frames using a TIRF microsopy setup described previously (*Lampert et al., 2010*). MetaMorph (Molecular Devices) software was used to compile images into videos and create kymographs out of individual moving microtubules. Velocity of gliding microtubules was calculated based upon the slope of the kymographs.

## Data analysis

The tracking of the proteins was performed automatically using the Definiens Software Suite. Prior to the analysis, the image data were processed performing a shading correction and smoothing. This was done by applying an 11 × 11 pixel (1463 × 1463 nm) kernel median filter and dividing the original raw image by the filtered image data. The resulting image was smoothed, using a 3 × 3 (399 × 399 nm) kernel Gaussian filter. The processed image data were searched and segmented for fluorescent signal. The signal area was searched for local intensity maxima within a search range of 532 nm. Circular objects with a radius of 332 nm were created on the found maxima and used for tracking the identified proteins.

The tracking of the movement of the TMR-, Alexa-488-, or GFP-labeled proteins was done through linking the proteins frame by frame by direct overlap, using the best fitting overlap.

## Bioinformatic methods

Cik1 protein sequences of *Saccharomyces cerevisiae* (accession number NP_013925.1), *Saccharomyces kudriavzevii* (EJT42048.1), *Saccharomyces arboricola* (EJS44163.1) and Vik1 sequences from *S. cerevisiae* (NP_015070.1), *S. kudriavzevii* (EJT43871.1), *S. arboricola* (EJS41496.1) were retrieved from the National Center for Biotechnology Information (NCBI). The *Saccharomyces bayanus* protein sequences of Cik1 (WashU_Sbay_Contig651.30) and Vik1 (WashU_Sbay_Contig637.18) were retrieved from the Saccharomyces Genome Database (SGD). The sequences were aligned with MAFFT (version 6, L-INS-I method) and further processed with Jalview and colored according to ClustalW.

## In vitro binding assay

Varying amounts (0.1–1 µM) of recombinant bait protein (GST, GST-Bim1$^{FL}$, GST-Bim1$^{0-185}$, GST-Bim1$^{185-344}$) were immobilized on 30 µl glutathione sepharose (GE Healthcare) in 0.5 ml binding buffer (25 mM Hepes pH 7.2, 250 mM NaCl, 1 mM MgCl$_2$, 1 mM EGTA, 0.5 mM DTT, 0.05% NP-40). The binding partner was added at a constant concentration between 0.5 and 1 µM and incubation lasted for 1 hr at 4°C. Afterwards beads were washed three times with 0.5 ml binding buffer and analyzed by SDS-PAGE and Coomassie Brilliant blue R250 staining.

## Yeast strains and spot assay

All modifications were performed in the S288C background (*Supplementary file 2*). Genetic modifications were introduced by using standard procedures.

For the spot assay, the desired strains were grown overnight in YPD medium. The following day cells were diluted to OD$_{600}$ = 0.6 which was the starting point of a 1:4 dilution series and spotted on YPD or 100 mM hydroxyurea (HU) plates. Plates were incubated at the indicated temperatures to 2–3 days.

## Live cell imaging

Imaging strains were grown in synthetic medium containing 2% glucose and imaged on concanavalin A-coated culture dishes (Matek) at ambient temperature. Eight z stacks with planes 0.3 µm apart were acquired at 30 s intervals on an Axiovert 200M microscope (Carl Zeiss) using an UPlanSApo 100× NA 1.40 oil immersion objective lens (Olympus) and a Coolsnap HQ CCD camera (Photometrics). Images were projected to two-dimensional images (SoftWoRx software) and further analyzed by MetaMorph (Molecular Devices). The linescans showing the fluorescence intensity for Kar3-3xGFP on spindles were plotted using ImageJ.

## Single-step affinity purification of native motors and mass spectrometry analysis

Desired strains were grown to OD$_{600}$ = 1.2 in YPD, centrifuged, drop frozen in liquid nitrogen, and lysed by freezer mill treatment. 5 g of yeast powder was dissolved in 10 ml buffer A (25 mM Hepes pH 8.0, 2 mM MgCl$_2$, 0.5 mM EGTA pH 8.0, 0.1 mM EDTA, 0.1% NP-40, 15% glycerole, 150 mM KCl, 0.01 mM ATP, 0.1 mM PMSF, 1× protease inhibitor cocktail set IV [Calbiochem]). The lysed cells were centrifuged twice, first at 43.146×g for 20 min and afterwards at 125.749×g for 1 hr. The resulting supernatant was incubated for 2–3 hr with 100 µl Dynabeads (Life Technologies) that were coupled to 50 µl anti-FLAG M2 antibody (Sigma Aldrich). Beads were washed three times with buffer A and four times with buffer B (25 mM Hepes pH 8.0, 150 mM KCl). Elution was achieved by using 1 beads volume 2 mg/ml 3xFLAG peptide in buffer B. The elution fractions were analyzed by SDS-PAGE and silver stain. For mass spectrometry analysis, an on-bead digest replaced the elution procedure: 500 ng LysC protease was added per 50 µl dynabeads in ammonium bicarbonate buffer and incubated at 37°C overnight. The supernatant was filtered and applied to mass spectrometry analysis, which was performed on three independent preparations.

## Acknowledgements

The authors thank all members of the Westermann lab for discussions and Jan-Michael Peters and David Keays for critical reading of the manuscript. We thank Karin Aumayr, Gabriele Stengl, and Pawel Paserbiek for help with microscopy and image analysis and Martin Colombini for fabrication of custom

mechanical components. We thank Marlene Brandstaetter and Alexander Schleiffer for excellent technical support. This work received funding from the European Research Council under the European Community's Seventh Framework Programme (SW FP7/2007-2013)/ERC grant agreement number [203499], from the Austrian Science Fund FWF (SW, SFB F34-B03) and the Austrian Research Promotion Agency (FFG). MIM acknowledges the VIPS Program of the Austrian Federal Ministry of Science and Research and the City of Vienna. The research leading to these results has received funding from the Vienna Science and Technology Fund (WWTF) project VRG10-11, Human Frontiers Science Program Project RGP0041/2012, Research Platform Quantum Phenomena and Nanoscale Biological Systems (QuNaBioS). The IMP is funded by Boehringer Ingelheim.

## Additional information

### Funding

| Funder | Grant reference number | Author |
|---|---|---|
| European Research Council (ERC) | FP2/2007-2013/no.203499 | Stefan Westermann |
| Austrian Science Fund (FWF) | SFB F34-B03 | Stefan Westermann |
| Wiener Wissenschafts-, Forschungs- und Technologiefonds | VRG10-11 | Alipasha Vaziri |
| Human Frontier Science Program (HFSP) | RGP004½012 | Alipasha Vaziri |

The funders had no role in study design, data collection and interpretation, or the decision to submit the work for publication.

### Author contributions

CM, MIM, SW, Conception and design, Acquisition of data, Analysis and interpretation of data, Drafting or revising the article; KD, BV, GL, Helped with single molecule in vitro imaging and in vivo experiments; GS, Helped with automated image analysis; AV, Conception and design, Analysis and interpretation of data, Drafting or revising the article

## Additional files

### Supplementary files

• Supplementary file 1. This file contains a detailed description of the mathematical model for cooperative Cik1–Kar3 kinesin movement.

• Supplementary file 2. This file contains tables listing the yeast strains and plasmids used in this study.

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
