## [Decision Letter]

Thank you for sending your work entitled “Non-catalytic motor domains enable processive movement and functional diversification of the kinesin Kar3” for consideration at *eLife*. Your article has been favorably evaluated by Randy Schekman (Senior editor), a Reviewing editor, and two reviewers.

The Reviewing editor and the reviewers discussed their comments before we reached this decision, and the Reviewing editor has assembled the following comments to help you prepare a revised submission.

The reviewers agreed that in principle this paper would be suitable for publication in *eLife*, but to make it so, the authors will need to carry out additional experiments and to reduce the strength of their claims, which both reviewers agreed were exaggerated (one reviewer even used the word “reckless”). The key result that the reviewers agreed makes the paper suitable for *eLife* is that the heterodimer walks processively toward the minus end of a MT.

1) The main conclusion of this manuscript is that Kar3 needs a second non-catalytic microtubule binding domain for processive translocation. If monomeric kinesin Kif1A can move along microtubules in vitro using a biased diffusion mechanism ([35], but see reservations below), Kar3 might do the same, obviating (or at least reducing) the requirement for a second motor head. A key question is whether the Cik1(1-360)-Kar3 mutant (or the Cik1(1-521)-Kar3 mutant) moves directionally in the presence of ATP. Now it is clear from Figure 2 that the velocity is substantially lower, but the authors need to analyze these data in more detail. If Cik1(1-360)-Kar3 and Cik1(1-521)-Kar3 do have significant directionality, then the authors might need to modify their description of the “foot-hold” mechanism.

2) Examples of exaggerated claims or incorrect information, all of which need to be fixed in a revision:

Abstract:

(i) “Whether this (hand-over-hand) model is universal, or if alternative mechanisms can support highly processive motion along microtubules, is an open question.” This statement is wrong. It is no longer an open question. Dewitt et al., Science, 2012 and Qiu et al., NSMB, 2012 showed that dynein walks in an uncoordinated manner.

(ii) “The noncatalytic domain acts as a foothold that allows Kar3 to bias translocation towards the minus end via a diffusive ADP state.” The reported experiments do not prove this statement. That kinesins have lower MT affinity in the ADP state was already known. The authors did not demonstrate how this motor steps along MT. In order to make these claims, they must demonstrate that: a) Cik1/Vik1 biases translocation of Kar3 to the minus end, b) translocation of Kar3 occurs in the ADP bound state and c) diffusion of the Kar3 head to the next tubulin binding site is a critical step. The authors need to delete this statement and replace it with one that sticks to the facts.

(iii) “This mechanism… allows transport of the Ndc80 kinetochore complex in vitro and is critical for Kar 3 function in vivo.” The authors show that Kar3 can bind and transport the isolated Ndc80 complex, not the Ndc80 complex as part of an assembled kinetochore. It has yet to be shown whether this in vitro transport reflects a property critical for in vivo function of Kar3. This latter statement is what they propose, but do not show, in this manuscript. The authors need to delete this statement and replace it with one that sticks to the facts.

Introduction section:

“In general motors equipped with only one catalytically active head fail to achieve long-range processive motility”. This is not true. The authors need to rewrite this paragraph citing: (i) the Okada et al. 2003 Kif1A paper showing processive single-headed motility; (ii) the Kaseda, Higuchi and Hirose (NSMB, 2003) and Thoresen et al. (Biochemistry, 2008) papers showed that processive motility of kinesin heterodimers with mutant motor domains that have lower catalytic activity; (iii) the [12] paper on cytoplasmic dynein showing that processive motility can occur when one of the heads in the dimer is replaced by a non-motor domain; and (iv) reports of processive motility with axonemal dynein. In other words, the authors need to remove overblown claims, discuss earlier results in a balanced way, and clearly and accurately indicate how their results advance the field.

The manuscript lacks evidence that their preps do not include aggregates. The authors could rule out aggregation by running gel filtration experiments, which are standard in the field.

The authors do cite evidence in Figure 1–figure supplement 1, for mostly single step photobleaching of labeled motors, arguing that processive motility is a property of individual motors. While this may be true, the single step photobleaching is not definitive. First, the data clearly show that some motors show 2 or >2 steps of photobleaching, suggesting protein aggregation or transient interactions. Second, the labeling efficiency is not calculated. If it is low (e.g. 40%), the fact that 10-20% of motors have 2 or more steps of photobleaching would suggest most proteins are aggregates. Third, if the motors are aggregated (e.g. 10 motors in an aggregate) then almost all the aggregates will have TMR even if the coupling efficiency is low. Aggregation may be an even bigger problem with the delta K-loop construct, which is expected to be less soluble. So gel filtration on this construct *is* also necessary.

In Figure 1, how do the authors know that the MTs are antiparallel and overlap? Was the polarity marked?

Third paragraph of the Results section:

“We noticed that due to their relatively large velocity distribution…” This whole paragraph does not make sense. How do they know that the motors run into each other (e.g. literally colliding with each other)? What is the evidence that motors travel together in teams? Are the proteins in contacts with each other (which cannot be seen by a fluorescence microscope due to diffraction limited resolution of the assay). In this section, they can also cite the Cin8 study by the Surrey lab (Roostalu et al., Science, 2011).

Eighth paragraph of the Results section:

The interpretation of the K-tail mutant experiments is counterintuitive. One would expect that removing positively charged residues from the noncatalytic head would reduce its affinity for MTs (which is not shown biochemically in this manuscript and needs to be shown in a revised version). Since the authors wish to refer to the function of this head as “a foothold”, these mutations would be expected to alter motility by reducing processivity (because it is no longer a very strong MT tether), with increased or unchanged velocity (because it might be easier for Kar3 to pull off the noncatalytic head from MT by force). The description of these results and their interpretation need reconsideration and revision.

The authors do not present direct evidence that Kar3 transports Ndc80 in vivo. Therefore, they need to change “We conclude that the Ndc80 complex is a direct cargo for Kar3 transport” to “We conclude that the Ndc80 complex can be transported by the Kar3/Cik1 motor.”

Discussion:

(i) In the first paragraph, the authors claim to have discovered that Kar3/Cik or Kar3/Vik1 heterodimers have a new mode of processive motility, and they assert that this is the first example of a motor that violates hand over hand motility. These statements are wrong (see comments above). Also note that without high resolution stepping studies their conclusions are too speculative.

(ii) The authors claim that “Kar3's motile characteristics share similarities to Kif1A”. How so? Kif1A was claimed to be processive on its own and not to require a foothold. The effect of the positive charges is different. Note that processive motility of single-headed Kif1 has not been replicated in another laboratory. The authors should analyze the motility of a Kar3 monomer and compare their results with those published by Hirokawa on Kif1A.

Figure 7:

The model shown does not relate to the main story, and there is no evidence provided for it. It should be removed from the manuscript. The entire discussion of diffusive states, anisotropy and powerstroke are far too speculative. It would be more appropriate for some future review. The problem is that without high-resolution stepping data many of the conclusions are in fact speculations. The authors have to be very careful to distinguish fact from fiction.

In summary, to be suitable for *eLife*, a revised manuscript will need to include at least the following:

1) More convincing and complete demonstration, by careful gel filtration experiments, that the motor proteins are not aggregated.

2) More thorough analysis of the data in Figure 2: is there directionality in the motion detected for the mutants?

3) Comparison of the behavior of a Kar3 monomer with the reported behavior of Kif1A.

The revised manuscript must also avoid the exaggerated and overstated claims that mar the current version.

---

## [Author Response]

*In summary, to be suitable for* eLife*, a revised manuscript will need to include at least the following*:

*1) More convincing and complete demonstration, by careful gel filtration experiments, that the motor proteins are not aggregated*.

*2) More thorough analysis of the data in*
Figure 2*: is there directionality in the motion detected for the mutants?*

3) Comparison of the behavior of a Kar3 monomer with the reported behavior of Kif1A.

*The revised manuscript must also avoid the exaggerated and overstated claims that mar the current version*.

1) In the revised manuscript we have included a biochemical characterization of Cik1-Kar3 motors by gel filtration and velocity gradient centrifugation analysis (new Figure 1). To go further and more directly characterize the motors we have visualized the motor proteins used in the experiments by low angle Platinum/Carbon rotary shadowing EM (new Figure 1). The hydrodynamic analysis shows that the motor proteins are biochemically well behaved, as they elute as a single species with no indication of aggregation. The EM data (the first visualization of full-length Cik1-Kar3 motors) reveals the overall organization of the motors and supports the conclusion that individual Cik1-Kar3 heterodimers are being analyzed.

2) As suggested we have more carefully analyzed the movement of wild-type and mutant motors in ATP in the new Figure 4. A mean-squared displacement analysis indicates that the movement of the Cik1(1-360)-Kar3 mutant is fitted well with n=1 to the one-dimensional diffusion equation <x^2^>= a·t^n^, indicating random diffusion with no directional component. In contrast, data for mean squared displacement of the wild-type motor is fitted with n=2, indicating directed movement.

3) We have constructed a monomeric, Halo-tagged Kar3 construct and imaged it in the TIRF assay. In the presence of ATP and under low nanomolar motor concentrations, only very short-lived binding events can be detected on microtubules (new Figure 4). Thus, there is no evidence that monomeric Kar3 can move processively.

We believe that taken together these three points have greatly strengthened the manuscript. The biochemical and EM characterization shows that well-behaved, non-aggregated motors were purified. The additional experiments show that indeed the non-catalytic head is strictly required for processive movement, as no directional movement of the heterodimer can be detected after its removal and in the reciprocal experiment; the Kar3 monomeric head alone does not display processive movement.

*The reviewers agreed that in principle this paper would be suitable for publication in* eLife*, but to make it so, the authors will need to carry out additional experiments and to reduce the strength of their claims, which both reviewers agreed were exaggerated (one reviewer even used the word “reckless”). The key result that the reviewers agreed makes the paper suitable for* eLife *is that the heterodimer walks processively toward the minus end of a MT*.

*1) The main conclusion of this manuscript is that Kar3 needs a second non-catalytic microtubule binding domain for processive translocation. If monomeric kinesin Kif1A can move along microtubules in vitro using a biased diffusion mechanism (*[35]*, but see reservations below), Kar3 might do the same, obviating (or at least reducing) the requirement for a second motor head. A key question is whether the Cik1(1-360)-Kar3 mutant (or the Cik1(1-521)-Kar3 mutant) moves directionally in the presence of ATP. Now it is clear from*
Figure 2
*that the velocity is substantially lower, but the authors need to analyze these data in more detail. If Cik1(1-360)-Kar3 and Cik1(1-521)-Kar3 do have significant directionality, then the authors might need to modify their description of the “foot-hold” mechanism*.

Please see our response to this point above. In addition to the analysis for the Cik1 (1-360)-Kar3 mutant presented in the new Figure 4, we have conducted the same analysis for the Cik1 (1-521)-Kar3 mutant (Figure 9).Author response image 1.n=1 for the fit indicates that the movement of this mutant also does not contain a directional component.

*2) Examples of exaggerated claims or incorrect information, all of which need to be fixed in a revision*:

*Abstract*:

*(i) “Whether this (hand-over-hand) model is universal, or if alternative mechanisms can support highly processive motion along microtubules, is an open question*.*”*

*This statement is wrong. It is no longer an open question. Dewitt et al., Science, 2012 and Qiu et al., NSMB, 2012 showed that dynein walks in an uncoordinated manner*.

As suggested we have rewritten the Abstract. We would like to point out, however, that we explicitly referred to kinesin motors in the original Abstract. Indeed, Dynein movement does not require head-to-head coordination and we have cited the above mentioned papers in our Discussion.

*(ii) “The noncatalytic domain acts as a foothold that allows Kar3 to bias translocation towards the minus end via a diffusive ADP state*.*”*

*The reported experiments do not prove this statement. That kinesins have lower MT affinity in the ADP state was already known. The authors did not demonstrate how this motor steps along MT. In order to make these claims, they must demonstrate that: a) Cik1/Vik1 biases translocation of Kar3 to the minus end, b) translocation of Kar3 occurs in the ADP bound state and c) diffusion of the Kar3 head to the next tubulin binding site is a critical step. The authors need to delete this statement and replace it with one that sticks to the facts*.

We have deleted and replaced the corresponding statements in the revised Abstract.

*(iii) “This mechanism… allows transport of the Ndc80 kinetochore complex in vitro and is critical for Kar 3 function in vivo.” The authors show that Kar3 can bind and transport the isolated Ndc80 complex, not the Ndc80 complex as part of an assembled kinetochore. It has yet to be shown whether this in vitro transport reflects a property critical for in vivo function of Kar3. This latter statement is what they propose, but do not show, in this manuscript. The authors need to delete this statement and replace it with one that sticks to the facts*.

We have replaced the statement with “… can support transport of the Ndc80 complex in vitro…”.

*Introduction section*:

*“In general motors equipped with only one catalytically active head fail to achieve long-range processive motility”. This is not true. The authors need to rewrite this paragraph citing: (i) the Okada et al. 2003 Kif1A paper showing processive single-headed motility; (ii) the Kaseda, Higuchi and Hirose (NSMB, 2003) and Thoresen et al. (Biochemistry, 2008) papers showed that processive motility of kinesin heterodimers with mutant motor domains that have lower catalytic activity; (iii) the*
[12]
*paper on cytoplasmic dynein showing that processive motility can occur when one of the heads in the dimer is replaced by a non-motor domain; and (iv) reports of processive motility with axonemal dynein. In other words, the authors need to remove overblown claims, discuss earlier results in a balanced way, and clearly and accurately indicate how their results advance the field*.

We have re-written the Introduction as suggested. In general we have put less emphasis on a comparison of a conventional hand-over-hand mechanism with the observed motility of Kar3 heterodimers and instead have focused on a comparison of Kar3 to other Kinesin-14 family members such as Ncd.

*The manuscript lacks evidence that their preps do not include aggregates. The authors could rule out aggregation by running gel filtration experiments, which are standard in the field*.

We have included these experiments in the revised manuscript (new Figure 1), please see our response to the three general points above.

*The authors do cite evidence in Figure 1–figure supplement 1, for mostly single step photobleaching of labeled motors, arguing that processive motility is a property of individual motors. While this may be true, the single step photobleaching is not definitive. First, the data clearly show that some motors show 2 or >2 steps of photobleaching, suggesting protein aggregation or transient interactions. Second, the labeling efficiency is not calculated. If it is low (e.g. 40%), the fact that 10-20% of motors have 2 or more steps of photobleaching would suggest most proteins are aggregates. Third, if the motors are aggregated (e.g. 10 motors in an aggregate) then almost all the aggregates will have TMR even if the coupling efficiency is low. Aggregation may be an even bigger problem with the delta K-loop construct, which is expected to be less soluble. So gel filtration on this construct* is *also necessary*.

We have estimated the TMR labeling efficiency using a construct that in addition to the Halo tag contains an eGFP fusion. By comparing eGFP to TMR fluorescence at low motor concentrations we estimate the labeling efficiency to exceed 90% (see fluorescence micrographs, Figure 10). Together with the photobleaching steps, the biochemical characterization of the oligomeric state, and the two-color motor mixing experiment, there is strong evidence that heterodimers are imaged. To the last point: we have removed the experiments with the delta K-tail construct in the revised manuscript.Author response image 2.Fluorescence Micrographs.

*In*
Figure 1, *how do the authors know that the MTs are antiparallel and overlap? Was the polarity marked?*

Bundles of microtubules are identified by increased fluorescence in the tubulin channel. In this experiment the polarity of the microtubules was not marked but inferred from the behavior of the motors.

*Third paragraph of the Results section*:

*“We noticed that due to their relatively large velocity distribution…” This whole paragraph does not make sense. How do they know that the motors run into each other (e.g. literally colliding with each other)? What is the evidence that motors travel together in teams? Are the proteins in contacts with each other (which cannot be seen by a fluorescence microscope due to diffraction limited resolution of the assay). In this section, they can also cite the Cin8 study by the Surrey lab (Roostalu et al., Science, 2011)*.

We have re-worded this section to state that we are simply comparing properties of motors with different brightness, i.e. heterodimers compared to teams that consist of two or more heterodimers

*Eighth paragraph of the Results section*:

The interpretation of the K-tail mutant experiments is counterintuitive. One would expect that removing positively charged residues from the noncatalytic head would reduce its affinity for MTs (which is not shown biochemically in this manuscript and needs to be shown in a revised version). Since the authors wish to refer to the function of this head as “a foothold”, these mutations would be expected to alter motility by reducing processivity (because it is no longer a very strong MT tether), with increased or unchanged velocity (because it might be easier for Kar3 to pull off the noncatalytic head from MT by force). The description of these results and their interpretation need reconsideration and revision.

While our experiments establish that the non-catalytic head in combination with Kar3 determines the velocity of the motor complex, we indeed at this point cannot give a mechanistic explanation for the role of the “K-tail”. In additional experiments we determined that simply adding the “K-tail” to the carboxyterminus of Cik1-Kar3 is not sufficient to alter the velocity of this motor. Thus, the K-tail’s role in the mechanochemical cycle might be more complex than simply altering the affinity for microtubules. This is in line with recent work from Yoshi et al., JBC, 2013. We have decided to remove the figure panels describing the K-tail experiments from the revised manuscript, as these points require further experimentation and do not contribute to the main message of the paper.

*The authors do not present direct evidence that Kar3 transports Ndc80 in vivo. Therefore, they need to change “We conclude that the Ndc80 complex is a direct cargo for Kar3 transport” to “We conclude that the Ndc80 complex can be transported by the Kar3/Cik1 motor*.*”*

We agree and have changed the wording accordingly.

*Discussion*:

*(i) In the first paragraph, the authors claim to have discovered that Kar3/Cik or Kar3/Vik1 heterodimers have a new mode of processive motility, and they assert that this is the first example of a motor that violates hand over hand motility. These statements are wrong (see comments above). Also note that without high resolution stepping studies their conclusions are too speculative*.

*(ii) The authors claim that “Kar3's motile characteristics share similarities to Kif1A”. How so? Kif1A was claimed to be processive on its own and not to require a foothold. The effect of the positive charges is different. Note that processive motility of single-headed Kif1 has not been replicated in another laboratory. The authors should analyze the motility of a Kar3 monomer and compare their results with those published by Hirokawa on Kif1A*.

As suggested, we have tested the motility of a monomeric Kar3 head construct (residues 353-729 fused to an N-terminal Halo tag). We have found no evidence for processive movement under conditions that allow robust motility of the heterodimer (new Figure 4). We agree that the suggested similarities to Kif1A have been confusing in our Discussion. We have rewritten the Discussion to conclude that Kar3’s mechanism of movement must be different from Kif1A as wehave not detected motility of a monomeric Kar3 construct and no directional movement was observed after removing the non-catalytic Cik1 head.

Figure 7:

*The model shown does not relate to the main story, and there is no evidence provided for it. It should be removed from the manuscript. The entire discussion of diffusive states, anisotropy and powerstroke are far too speculative. It would be more appropriate for some future review. The problem is that without high-resolution stepping data many of the conclusions are in fact speculations. The authors have to be very careful to distinguish fact from fiction*.

As suggested, we have re-written the discussion section to remove the more speculative statements and focus on the experimentally supported insights. We have removed the original Figure 7, as we can indeed not give conclusive insights about the stepping mechanism. We would like to maintain a simplified model (new Figure 8) that illustrates the following experimentally addressed points: 1) the Kar3 motor can take multiple steps along the microtubule before dissociating; 2) the non-catalytic head is required for the movement by promoting tight binding in no-nucleotide and AMP-PNP state 3) the movement likely has a diffusional component, as suggested by the diffusive behavior of the motor in ADP and the pronounced salt-sensitivity of the velocity.

We agree that further biophysical experiments are required to establish a mechanism and we have greatly tightened the discussion to avoid too much speculation about the mechanism.